# Gut mucin fucosylation dictates the entry of botulinum toxin complexes

Sho Amatsu[1,2,16], Takuhiro Matsumura[1,16], Chiyono Morimoto[1], Sunanda Keisham[3,4], Yoshiyuki Goto[5,6,7,8], Tomoko Kohda[9], Jun Hirabayashi[10], Kengo Kitadokoro[11], Takane Katayama[12], Hiroshi Kiyono [8,13,14,15], Hiroaki Tateno [3,4], Masahiko Zuka[2] & Yukako Fujinaga [1] ✉

Botulinum toxins (BoNTs) produced by *Clostridium botulinum* are the most potent known bacterial toxins. The BoNT complex from serotype B-Okra (LPTC/B^Okra) exerts at least 80-fold higher oral toxicity in mice compared with that from serotype A1 (L-PTC/A^62A). Here, we show that L-PTC/B^Okra is predominantly absorbed through enterocytes, whereas LPTC/A^62A targets intestinal microfold cells. Furthermore, α1,2-fucosylation of intestinal mucin determines the oral toxicity of L-PTCs as well as their entry routes, due to differential carbohydrate-binding spectrum of one of the L-PTC components, the hemagglutinin (HA) complex. Fucosylation-deficient mice display reduced intestinal mucin penetration of L-PTC/B^Okra via HA, and lower susceptibility to oral intoxication with this toxin. Thus, our results shed light on the molecular mechanisms by which the oral toxicity of BoNTs is increased after crossing intestinal mucus layers

*Clostridium botulinum* is the causative pathogen of botulism, which can result in neuromuscular paralysis and death[1]. Botulism is caused by botulinum toxin (BoNT) complexes comprising large and medium progenitor toxin complexes (L-PTCs and M-PTCs, respectively)[2]. M-PTCs consist of BoNT and non-toxic non-hemagglutinin (NTNHA), which can be further combined with hemagglutinin (HA) to form L-PTCs or with OrfX2 protein to form M-PTC−OrfX2 complex. BoNT poisoning often occurs through ingestion of food contaminated with *C. botulinum*. After ingestion, BoNT is protected within M-PTCs to avoid destruction in the hostile environment of the gastrointestinal (GI) tract[3,4], and is absorbed from the small intestine into the bloodstream[2]. HA is a large protein complex which comprises three subcomponents: HA1, HA2, and HA3 (also known as HA33, HA17, and HA70, respectively)[5]. This complex facilitates BoNT transcytosis and translocation across the intestinal epithelium through the carbohydrate-binding activity[6–8] and the E-cadherin-binding activity[6,7,9,10], respectively. Recently, OrfX has been reported to facilitate the intestinal absorption of

---

[1]Department of Bacteriology, Graduate School of Medical Sciences, Kanazawa University, Ishikawa, Japan. [2]Department of Forensic Medicine and Pathology, Graduate School of Medical Sciences, Kanazawa University, Ishikawa, Japan. [3]Cellular and Molecular Biotechnology Research Institute, National Institute of Advanced Industrial Science and Technology (AIST), Tsukuba, Japan. [4]Ph.D. Program in Human Biology, School of Integrative and Global Majors, University of Tsukuba, Tsukuba, Japan. [5]Division of Molecular Immunology, Medical Mycology Research Center, Chiba University, Chiba, Japan. [6]Division of Pandemic and Post-disaster Infectious Diseases, Research Institute of Disaster Medicine, Chiba University, Chiba, Japan. [7]Division of Infectious Disease Vaccine R&D, Research Institute of Disaster Medicine, Chiba University, Chiba, Japan. [8]Chiba University Synergy Institute for Futuristic Mucosal Vaccine Research and Development (cSIMVa), Chiba University, Chiba, Japan. [9]Graduate School of Veterinary Sciences, Osaka Metropolitan University, Osaka, Japan. [10]Institute for Glyco-core Research (iGCORE), Nagoya University, Nagoya, Japan. [11]Faculty of Molecular Chemistry and Engineering, Graduate School of Science and Technology, Kyoto Institute of Technology, Kyoto, Japan. [12]Graduate School of Biostudies, Kyoto University, Kyoto, Japan. [13]Department of Human Mucosal Vaccinology, Chiba University Hospital, Inohana, Chuo-ku, Chiba, Japan. [14]Department of Medicine, School of Medicine and Chiba University-University of California San Diego Center for Mucosal Immunology, Allergy and Vaccine (CU-UCSD cMAV), University of California, San Diego, CA, USA. [15]Future Medicine Education and Research Organization, Chiba University, Inohana, Chuo-ku, Chiba, Japan. [16]These authors contributed equally: Sho Amatsu, Takuhiro Matsumura. ✉e-mail: yukafuji@med.kanazawa-u.ac.jp

BoNT by unknown functions[11]. Once in the circulation, BoNT is transported to the neuromuscular junction, where it blocks neurotransmitter release.

BoNTs have been classically categorized into seven serotypes (A–G), each with multiple subtypes[2,12]. Serotypes A, B, E, and rarely F are responsible for most natural human botulism. These toxins have similar intraperitoneal median lethal dose ($LD_{50}$) values in mice (e.g., BoNT serotype A1: 0.11–0.45 ng/kg[13], serotype B1: 0.21–0.50 ng/kg[13], serotype E1: 0.65–0.84 ng/kg[13], serotype F1: 3.6 ng/kg[14]) (Fig. 1a), whereas the L-PTC from serotype B1-Okra (L-PTC/B$^{Okra}$, termed hyper-oral-toxic) has 20–80-fold[2] (Fig. 1a) higher oral toxicity than that from serotype A1-62A (L-PTC/A$^{62A}$, termed non-hyper-oral-toxic). *C. botulinum* serotypes E and F produce only M-PTCs[2]. The molecular mechanism by which L-PTC/B$^{Okra}$ exerts high oral toxicity is unknown. Here, we show that the mucin fucosylation determines the entry routes of botulinum toxin complexes in the gut and their oral toxicities.

## Results

### Hyper-oral-toxic BoNT passes the mucus barrier and enters the host through enterocytes

To enter the host, L-PTC/A$^{62A}$ targets microfold (M) cells within the follicle-associated epithelium (FAE) of Peyer's patches[8] (Fig. 1b). We found that L-PTC/B$^{Okra}$ bound to enterocytes within the villous epithelium (VE) and FAE and underwent endocytosis (Fig. 1b and Supplementary Fig. 1). L-PTC/B$^{Okra}$ also bound to M cells (Fig. 1b). These results indicate that L-PTC/B$^{Okra}$ is absorbed via a different intestinal route than L-PTC/A$^{62A}$. In the small intestine, the enterocytes make up more than 80% of total intestinal epithelial cells[15], whereas the number of M cells is very limited; less than one in $10^7$ of total intestinal epithelial cells[16]. We hypothesized that numerous entry points contribute to the higher oral toxicity of L-PTC/B$^{Okra}$.

The epithelium is maintained by a gut barrier consisting of a protective mucus layer[17]. The mucus, which is composed of gel-forming mucins, forms a matrix of gels and functions as a physical

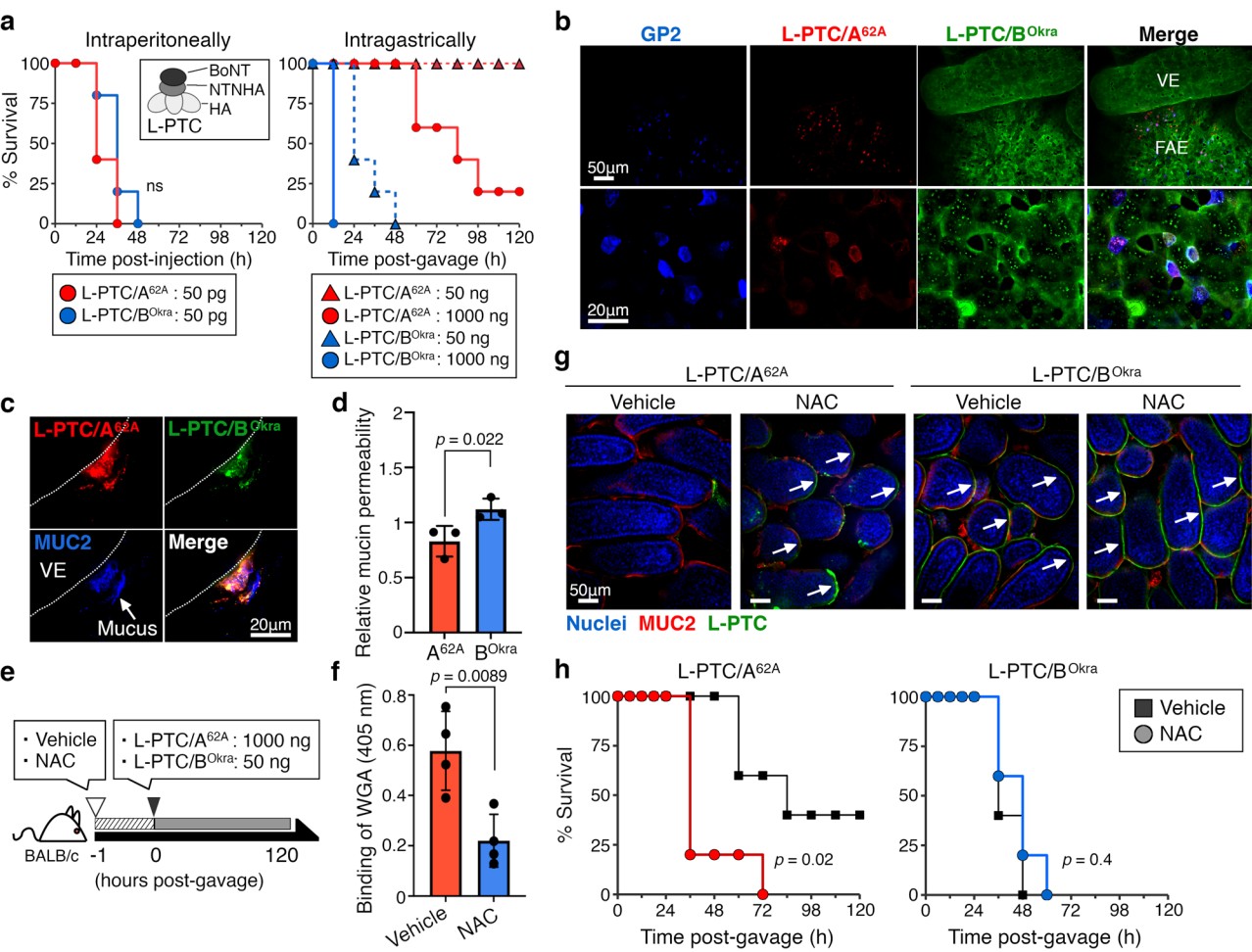

**Fig. 1 | A hyper-oral-toxic BoNT complex from Clostridium botulinum strain B-Okra passes the mucus barrier and is absorbed through enterocytes in the small intestine. a** Survival curves of BALB/c mice ($n = 5$ per group) administered large progenitor toxin complexes (L-PTC/A$^{62A}$ or L-PTC/B$^{Okra}$) intraperitoneally (i.p.; 50 pg) or intragastrically (i.g.; 50 or 1000 ng). **b, c** Representative images of whole-mounted small intestine and L-PTCs. A mixture of Alexa Fluor (AF) 568–labeled L-PTC/A$^{62A}$ (red) and AF 488–labeled L-PTC/B$^{Okra}$ (green) was injected into ligated mouse intestinal loops. M cells (**b**) and mucin (**c**) were visualized with anti-glycoprotein 2 (GP2)[16] and anti-MUC2 antibodies, respectively (blue). VE villous epithelium, FAE follicle-associated epithelium. Scale bars, 50 μm (low magnification) in upper panels of **b, g**; 20 μm (high magnification) in lower panels of **b, c**. **d** Mucin penetration of L-PTCs was quantified using transwell inserts coated with porcine gastric mucin (PGM). Upper-to-lower penetration was analyzed by

immunoblotting for L-PTCs and normalized to the bovine serum albumin (BSA) control. **e** Schematic representation of a mucus-depleted mouse model. Mice were gavaged with vehicle (sterile water) or 100 mg/mL *N*-acetylcysteine (NAC) 1 h before the in situ loop assay and challenge. **f** Small intestinal mucins were collected from vehicle- or NAC-treated mice. To quantify the mucin concentration, binding of wheat germ agglutinin (WGA) to the mucin was analyzed by ELISA. **g** Binding of L-PTCs (green) to the villous epithelium. Nuclei and mucin were visualized with Hoechst 33342 (blue) and an anti-MUC2 antibody (red). Arrows indicate L-PTC attachment to the epithelium. **h** NAC-treated mice ($n = 5$ per group) were challenged with L-PTC/A$^{62A}$ (1000 ng) and L-PTC/B$^{Okra}$ (50 ng). **d, f** Values represent the mean ± SD of triplicate (**d**) and four independent samples (**f**), two-tailed Student's *t*-test. **a, h** Log-rank test.

barrier, trapping pathogens and noxious agents. We found that both of the studied L-PTCs were trapped in the MUC2 mucin of the mucus layer (Fig. 1c and Supplementary Fig. 1). L-PTC/B$^{Okra}$ penetrated the mucin to a greater extent than L-PTC/A$^{62A}$ (Fig. 1d and Supplementary Fig. 2). L-PTC/B$^{Okra}$ bound to the luminal surface of the villous epithelium and underwent endocytosis (Fig. 1c and Supplementary Fig. 1). To facilitate the clearance of the mucus layer, we treated mice with $N$-acetylcysteine (NAC), which liquefies mucus by reducing the disulfide bonds in mucus proteins and inhibits mucin synthesis[18] (Fig. 1e,f). The NAC treatment allowed L-PTC/A$^{62A}$ to bind to enterocytes within the villous epithelium, as L-PTC/B$^{Okra}$ did (Fig. 1g). Moreover, NAC-treated mice were more susceptible than control mice to toxicity following oral administration of L-PTC/A$^{62A}$, although NAC did not alter the toxicity of L-PTC/B$^{Okra}$ (Fig. 1h). These findings suggest that L-PTC/B$^{Okra}$ penetrates the mucus layer without interacting with mucin and enters the host through enterocytes, leading to the higher oral toxicity.

## Mucus trapping is dependent on HA's carbohydrate-binding activities

To explore toxin–mucin interactions, we performed ELISA of porcine gastric mucin (PGM). We found that L-PTC/A$^{62A}$ bound strongly to PGM, whereas L-PTC/B$^{Okra}$ barely did so (Fig. 2a). Thus, the binding of L-PTCs to PGM was inversely related to their mucin permeability (Fig. 1d). The M-PTC from neither serotype bound to PGM (Fig. 2a). These results indicate that the interactions between L-PTCs and mucins are dependent on HAs. Consistent with this, HA alone bound to PGM and localized in the small intestine, as did each L-PTC (Fig. 2a–c). These findings suggest that the entry routes of BoNTs are dictated by HAs. To test this, we generated a recombinant chimeric L-PTC (rL-PTC/BA) composed of BoNT/B$^{Okra}$, NTNHA of serotype B1-Okra (NTNHA/B$^{Okra}$), and HA of serotype A1-62A (HA/A$^{62A}$) (Fig. 2d)[19]. rL-PTC/BA exhibited reduced oral toxicity despite its comparable intraperitoneal toxicity (Fig. 2e). Although these rL-PTCs showed 4-fold decreased intragastric toxicities compared to the native L-PTC/B (Figs. 1a and 2e), we confirmed that each rL-PTC carries equivalent BoNT activity (Fig. 2e) and retains intact HA activities[19]: carbohydrate-binding and E-cadherin-binding. Collectively, the oral toxicities of L-PTCs were dependent on differences in HA rather than in BoNT or NTNHA.

In serotypes A and B, HA has two different carbohydrate-binding activities with the different subcomponents, HA1 and HA3[20] (Fig. 2f): HA1 recognizes a galactose (Gal) residue (lactose, A$^{62A}$: $K_D$ ~3.6 mM, B$^{Okra}$: $K_D$ ~1.6 mM)[21], while HA3 recognizes a sialic acid (Sia/$N$-acetylneuraminic acid (Neu5Ac)) residue (A$^{62A}$: $K_D$ ~7.8 mM)[22]. The HA/A$^{62A}$ and HA of serotype B1-Okra (HA/B$^{Okra}$) subcomponents have high sequence similarity (HA1/A$^{62A}$ $vs.$ HA1/B$^{Okra}$: ~84%, HA3/A$^{62A}$ $vs.$ HA3/B$^{Okra}$: ~98%) (Supplementary Fig. 3a, c) and similar carbohydrate-binding pockets[21] (Fig. 2f and Supplementary Fig. 4). The pockets, however, possess a few different amino acids, e.g., His281$^{62A}$/Asn282$^{Okra}$ in HA1 and Arg619$^{62A}$/Lys619$^{Okra}$ in HA3 (Fig. 2f). A-62A with the substitutions of these amino acids, primarily in HA1, showed the PGM binding affinity (Fig. 2g) and intestinal localization (Fig. 2h) similar to B-Okra, and vice versa. The mutations in HA3 had much smaller impact on binding ability than HA1, although there were still notable changes observed (WT vs. Arg/Lys 619 mutants; $p = 0.057$ for A-62A, $p = 0.037$ for B-Okra) (Fig. 2g). These findings suggest that these HA1 and HA3 amino acids (His/Asn$^{HA1-281/282}$ and Arg/Lys$^{HA3-619}$, respectively) can be used to classify the oral toxicity of these toxins. We compared the amino acid sequences of HA proteins collected from the NCBI database ($n = 423$), including 370 sequences of serotypes A and B. Multiple sequence alignment revealed that HA can be classified into three sequence types: B-Okra type (Asn$^{HA1-281/282}$/Lys$^{HA3-619}$, 11.6%), A-62A type (His$^{HA1-281/282}$/Arg$^{HA3-619}$, 60.8%), and a hybrid type (Asn$^{HA1-281/282}$/Arg$^{HA3-619}$, 27.6%) (Supplementary Fig. 3d). Based on the oral toxicities of three types of L-PTC (B-Okra, Asn$^{HA1-282}$/Lys$^{HA3-619}$; A-62A, His$^{HA1-281}$/Arg$^{HA3-619}$; Osaka05, Asn$^{HA1-280}$/Arg$^{HA3-619}$) (Supplementary Fig. 3e), these distinct

sequence types of HA could imply the potential oral toxicity of L-PTCs, classified as hyper-oral-toxic, non-hyper-oral-toxic and intermediate. Further epidemiological studies focusing on these three HA types will help determine their relationship to human foodborne botulism.

## Terminal α1,2-fucosylation is pivotal for HA binding to mucin

Mucins are densely $O$-glycosylated proteins in which the $O$-glycans contain core structures consisting of galactose, $N$-acetylgalactosamine (GalNAc), and $N$-acetylglucosamine (GlcNAc)[23]. The core structures can be terminally decorated with galactose, GalNAc, sialic acid, or fucose residues. The carbohydrate-binding activities are crucial for the effective interaction between mucin and HA of both A-62A and B-Okra (Fig. 2g). We hypothesized that mucin glycosylation would affect the carbohydrate-binding activities of HA. To test this, we further characterized the carbohydrate-binding activities of HA/A$^{62A}$ and HA/B$^{Okra}$ by inhibition/competition ELISA with carbohydrates/lectins (Supplementary Fig. 5) and glycan microarrays (Supplementary Data 1). We found that HA1/A$^{62A}$ bound to fucose-α(1,2)-galactose and GalNAc residues in addition to galactose, although HA1/B$^{Okra}$ bound only to galactose (Fig. 3a, c and Supplementary Fig. 6). In the competition ELISA, $Ulex\ europaeus$ agglutinin I (UEA-I), which recognizes the fucose-α(1,2)-galactose-β(1,4)-GlcNAc linkage[24], blocked the PGM binding of HA/A$^{62A}$ but not HA/B$^{Okra}$ (Fig. 3b, d). Peanut agglutinin (PNA), which recognizes a galactose-β(1,3)-GalNAc residue[24], blocked the binding of HA/B$^{Okra}$ (Fig. 3b, d). These findings indicate that the binding differences between HA/A$^{62A}$ and HA/B$^{Okra}$ arise from α1,2-fucosylation of the terminal galactose in PGM.

According to structural models of HA1 in complex with carbohydrates[21,22] (Fig. 3e and Supplementary Fig. 7), α1,2-fucosylation of lactose (Lac) in α1,2-fucosyllactose (2FL) is extended toward the His281$^{A-62A}$/Asn282$^{B-Okra}$ residues, suggesting that the Asn282$^{B-Okra}$ does not accommodate the fucose-extension. To confirm whether this fucose extension affects the HA1–galactose interaction, we assessed the binding ability of HAs. The binding of HA/A$^{62A}$ was inhibited by lactose and α1,2-fucosyllactose with similar half-inhibitory dose (IC$_{50}$) values (5.4 mM $vs.$ 5.9 mM, respectively) (Fig. 3c). Moreover, HA1/A$^{62A}$ recognized lactose and α1,2-fucosyllactose at similar binding affinity ($K_D$) values (-4.7 mM vs. -5.5 mM, respectively) (Fig. 3f and Supplementary Fig. 8). This indicates that fucosylation neither facilitates nor impairs HA1/A$^{62A}$ binding to glycans. By contrast, the α1,2-fucosylation increased the $K_D$ of HA1/B$^{Okra}$ 2.4-fold compared with lactose (-16 mM vs. -6.6 mM, respectively) (Fig. 3f and Supplementary Fig. 8), resulting in a 32-fold higher IC$_{50}$ of α1,2-fucosyllactose against HA/B$^{Okra}$ than lactose (121 mM vs. 3.8 mM, respectively) due to the multivalency effect[25] (Fig. 3c). These results suggest that His281 in the galactose-binding pocket of HA1/A$^{62A}$ accepts the α1,2-fucosylated extension of galactose, but Asn282 in the galactose-binding pocket of HA1/B$^{Okra}$ does not.

To investigate how terminal glycosylation of mucin influences the HA binding, we used α1,2-fucosidase from $Bifidobacterium\ longum$ (AfcA)[26] and α-$N$-acetylgalactosaminidase (NAGA) to remove the terminal fucose-α(1,2) and GalNAc-α linkage from mucin, respectively. Terminal fucose and GalNAc removal was verified using UEA-I and $Dolichos\ biflorus$ agglutinin (DBA), the latter of which recognizes an α-GalNAc residue[24] (Fig. 3g, h and Supplementary Fig. 9). We found that AfcA treatment significantly increased the binding of HA/B$^{Okra}$ to PGM but did not affect that of HA/A$^{62A}$ (Fig. 3g). NAGA did not affect the binding of either HA (Fig. 3h). These results confirm that terminal α1,2-fucosylation, not GalNAcylation, is responsible for the differences in binding abilities of HAs to PGM.

## Mucin fucosylation confers the oral toxicity of the hyper-oral-toxic toxin

Secreted mucins are found in the mucus that covers epithelial tissues, and their expression is relatively tissue specific[23,27]; e.g., MUC2 is found

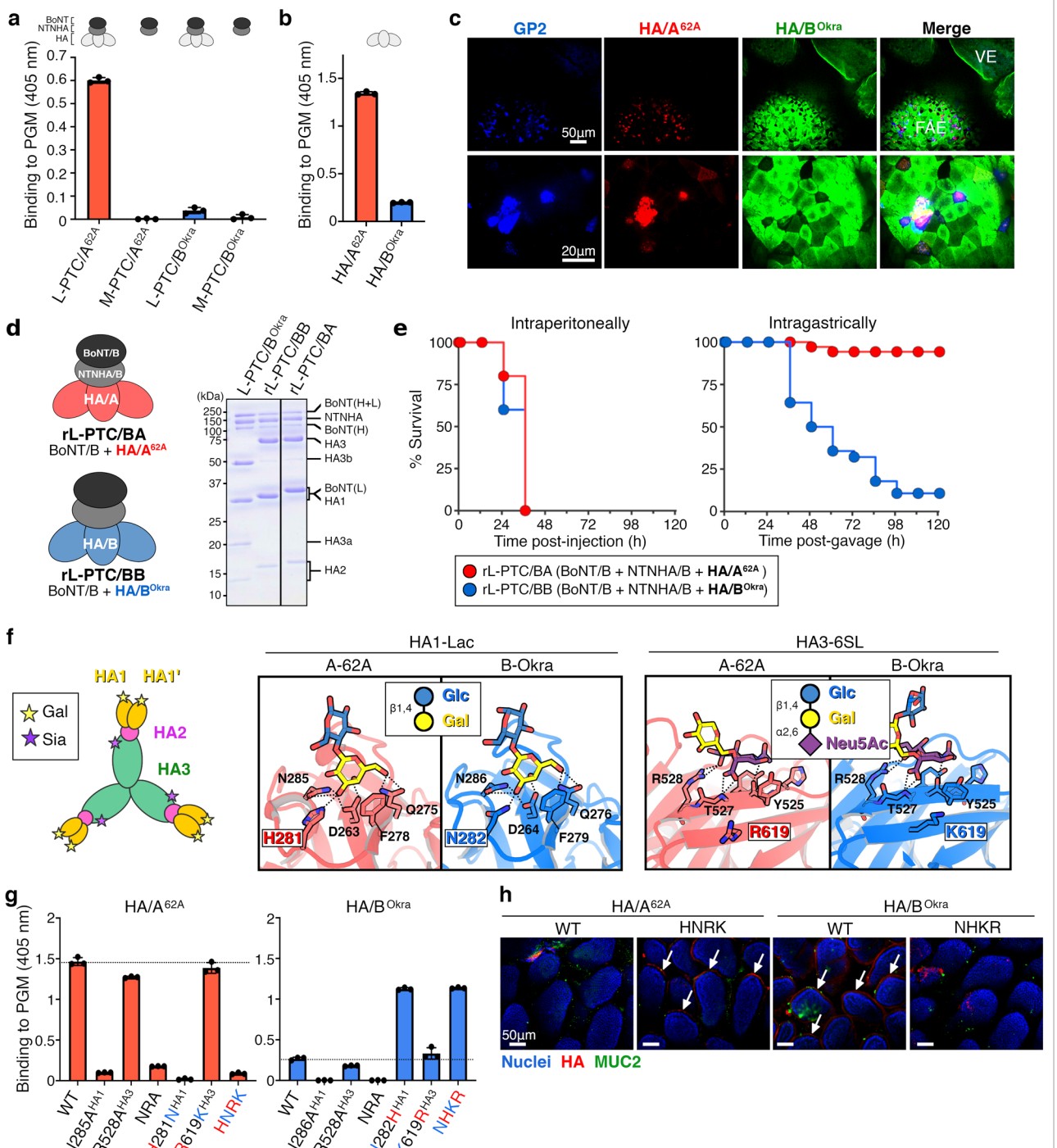

**Fig. 2 | Carbohydrate-binding activities of HA are pivotal for L-PTC binding and mucin penetration. a, b** Binding of PTCs (L-PTC, BoNT+NTNHA + HA; M-PTC, BoNT+NTNHA) (**a**) or HAs (**b**) to porcine gastric mucin (PGM) was analyzed by ELISA. **c** Representative images of whole-mounted small intestine with AF 568–labeled HA/A[62A] (red) and AF 488–labeled HA/B[Okra] (green). M cells were visualized with anti-GP2 antibody (blue). Scale bars, 50 μm (upper panels) and 20 μm (lower panels). **d** Chimeric recombinant L-PTCs (rL-PTCs) were reconstituted from BoNT/B[Okra], NTNHA/B[Okra], and HA (BB for HA/B[Okra], BA for HA/A[62A]). Purified proteins were verified by SDS-PAGE with Coomassie blue staining. **e** Mice were challenged with the rL-PTC/BA (red) or rL-PTC/BB (blue) via intraperitoneal injection (i.p., 100 pg; *n* = 10 per group) or intragastric administration (i.g., 200 ng; *n* = 35 for rL-PTC/BA and *n* = 28 for rL-PTC/BB). **f** Carbohydrate-binding sites of HA

(HA1/A[62A]-Lac: PDB ID 4LO2[22]; HA1/B[Okra]-Lac: 4OUJ[21]; HA3/A[62A]-6SL: 4LO5[22]; HA3/B[Okra]-6SL: 9UG6, this work). Ligands and interacting amino acids are shown in stick models colored light red (HA/A[62A]), light blue (HA/B[Okra]), yellow (galactose, Gal), blue (glucose, Glc), and purple (Neu5Ac). The omit map of the HA3/B[Okra]-6SL structure is shown in Supplementary Fig. 4. **g** ELISA was used to analyze the binding of mutated HAs to PGM. N285/6A[HA1] and R528A[HA3] are galactose- and sialic acid-binding–defective mutants, respectively[20]. NRA, N285/6A[HA1]/R528A[HA3]. HNRK, H281N[HA1]/R619K[HA3]. NHKR, N282H[HA1]/K619R[HA3]. **h** Ligated mouse intestinal loop assay with mutated HAs (red). Nuclei and mucin were visualized with Hoechst 33342 (blue) and an anti-MUC2 antibody (green). Arrows indicate HA attachment to the epithelium. Scale bars, 50 μm. **a, b, g** Values represent the mean ± SD of triplicate wells.

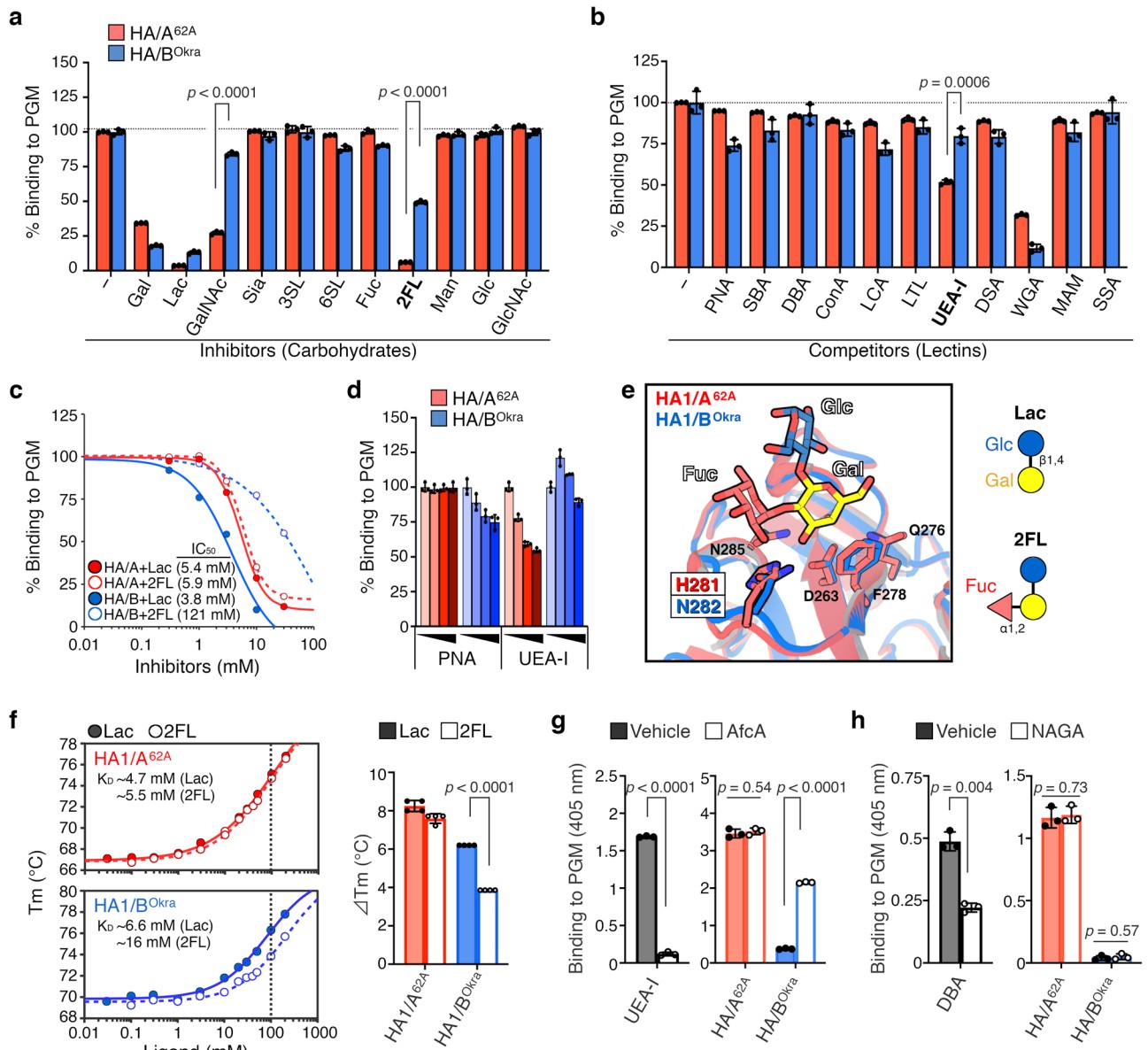

**Fig. 3 | Identification of a mucin-binding specificity determinant in HA. a–d,** binding of HAs to PGM was characterized by inhibition/competition ELISA with carbohydrates (**a, c**) or lectins (**b, d**). **a** The HA/A$^{62A}$ (red) or HA/B$^{Okra}$ (blue) were pre-incubated with carbohydrates (Gal galactose, Lac lactose, GalNAc *N*-acetylgalactosamine, Sia sialic acid, 3SL α2,3-sialyllactose, 6SL α2,6-sialyllactose, Fuc fucose, 2FL α1,2-fucosyllactose, Man mannose, Glc glucose, GlcNAc *N*-acetylglucosamine. **b** The PGM-coated plates were pre-treated with lectins (PNA peanut agglutinin, SBA soybean agglutinin, DBA *Dolichos biflorus* agglutinin, ConA concanavalin A, LCA *lens culinaris* agglutinin, LTL *lotus tetragonolobus* agglutinin, UEA-I *ulex europaeus* agglutinin I, DSA *Datura stramonium* agglutinin, WGA wheat germ agglutinin, MAM *Maackia amurensis* mitogen, SSA *Sambucus sieboldiana* agglutinin). The structure of carbohydrates and the specificity of lectins are described in Supplementary Fig. 5. **c** Half-maximal inhibitory concentration (IC$_{50}$) values of lactose (Gal-β1,4-Glc; filled circle) and α1,2-fucosyllactose (Fuc-α1,2-Gal-β1,4-Glc; open circle) against HA/A$^{62A}$ (red) and HA/B$^{Okra}$ (blue). **d** The binding of

HAs/A$^{62A}$ (red) or HA/B$^{Okra}$ (blue) was blocked by different concentrations of PNA (Gal-β1,3-GalNAc–specific lectin) and UEA-I (Fuc-α1,2-Gal–specific lectin). **e** Superimposition of HA1s in complex with α1,2-fucosyllactose (2FL). Crystal structures of HA1/A$^{62A}$-Lac (PDB ID 4LO2[22]) and HA1/B$^{Okra}$-Lac (PDB ID 4OUJ[21]) are superimposed with 2FL. **f** Melting temperature ($T_m$) values of HA1/A$^{62A}$ (red) and HA1/B$^{Okra}$ (blue) in the presence of lactose (filled circle) or α1,2-fucosyllactose (open circle) by thermal shift assay (left panel). $T_m$-shift ($\Delta T_m$) values were determined in the presence of ligands at a concentration of 100 mM (right panel) in quadruplicate. **g, h** PGM was treated with α1,2-fucosidase from *Bifidobacterium longum* (AfcA) (**g**) and α-*N*-acetylgalactosaminidase (NAGA) (**h**). The removal of the terminal fucose and GalNAc residues from mucin was detected with UEA-I and DBA (GalNAc-α–specific lectin), respectively. The binding of HAs to PGM treated with vehicle (filled) or enzymes (open) was analyzed with ELISA. Values represent the mean ± SD of triplicate (**a, b, c, d, g, h**) or quadruplicate (**f**) wells. The data were analyzed by a two-tailed Student's *t*-test.

in small intestine and colon, MUC5AC in stomach and colon, MUC5B in saliva. The glycosylation of these mucins is enormously diverse, and can vary between tissues, between individuals, and even within mucins from a single individual[28]. To examine the interaction between HA and intestinal mucin, we isolated mouse intestinal mucin (MIM) from the mucus of the upper small intestine, which is the main location for

absorption of orally ingested BoNTs[8]. HA/A$^{62A}$ had a higher affinity for MIM than HA/B$^{Okra}$ (Fig. 4a), and AfcA significantly increased the binding of HA/B$^{Okra}$ to MIM (Fig. 4a). These results confirm that the binding of HAs to MIM is largely consistent with their binding to PGM.

Fucosyltransferase 2 (FUT2) is one of the enzymes responsible for adding fucose to proteins or lipids by α1,2-fucosylation within the GI

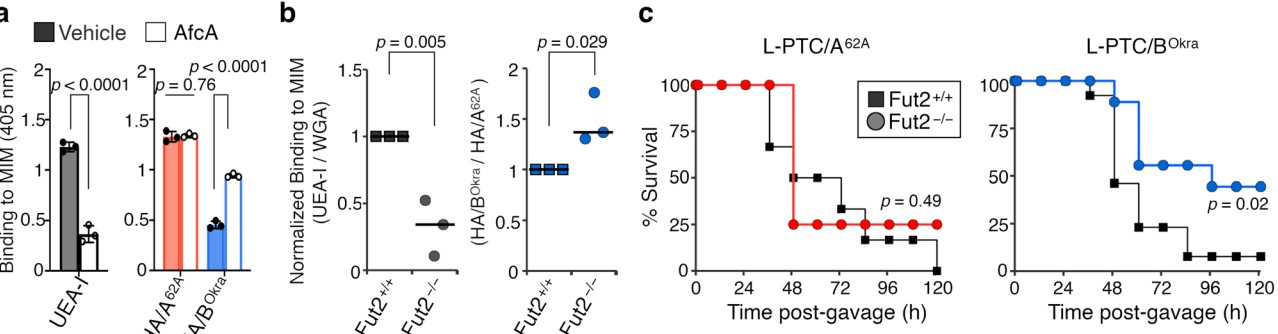

**Fig. 4 | L-PTC/BOkra passes through α1,2-fucosylated mucin and has high oral toxicity in mice. a** Binding of UEA-I (Fuc-α1,2-Gal−specific lectin) or HAs to mouse intestinal mucin (MIM) isolated from BALB/c mice. The MIM-coated plates were pre-treated with vehicle (filled) or AfcA (open). **b** The binding of lectins (left panel) and HAs (right panel) to MIM isolated from C57BL/6 J $Fut2^{+/+}$ or $Fut2^{-/-}$ mice. The binding of UEA-I or HA/B$^{Okra}$ were normalized to those of WGA (Sia− and GlcNAc−specific lectin) and HA/A$^{62A}$, respectively. **c** WT (square) and $Fut2^{-/-}$ (circle) mice were challenged i.g. with L-PTC/A$^{62A}$ (2 μg, $n = 6$ for WT and $n = 4$ for $Fut2^{-/-}$) and L-PTC/B$^{Okra}$ (3 ng, $n = 13$ for WT and $n = 9$ for $Fut2^{-/-}$). **a, b** Values represent the mean ± SD of triplicate wells (**a**) and the median of three independent samples (**b**). **a, b** Two-tailed Student's $t$-test. **c** Log-rank test.

tract[29]. FUT2 expression is observed in epithelial cells, including goblet cells, and is involved in the synthesis of histo-blood group antigens. Indeed, intestinal mucin isolated from $Fut2$-null ($Fut2^{-/-}$) mice (MIM$^{Fut2}$) showed low α1,2-fucosylation compared with WT ($Fut2^{+/+}$) mice (MIM$^{WT}$) (Fig. 4b and Supplementary Fig. 10). HA/B$^{Okra}$ had higher affinity for MIM$^{Fut2}$ than MIM$^{WT}$ (Fig. 4b; $n = 3$). Given these HA−MIM interactions, we determined the physiological relevance of the interaction between HA and mucin fucosylation to the oral toxicity of L-PTCs in vivo. $Fut2$-null mice were significantly resistant to the toxicity of orally administered L-PTC/B$^{Okra}$, but not to L-PTC/A$^{62A}$ (Fig. 4c). Taken together, these findings demonstrate that α1,2-fucosylation of intestinal mucin determines the entry routes and oral toxicities of L-PTCs by the carbohydrate-binding spectrum of HA.

## Discussion

Fucose is one of the major components of glycoproteins such as $N$-glycans and mucin-type $O$-glycans[30,31]. Fucosylation typically involves terminal modifications, resulting for instance in α1,2-, α1,3-, α1,4-, or α1,6-linked fucose. FUT2-mediated intestinal α1,2-fucosylation is abundant in the gut and is vital for host−microbe interactions and for maintaining gut homeostasis[32]. Commensal bacteria, e.g., *Bifidobacteria*, *Bacteroides*, and *Akkermansia muciniphila*, cleave the terminal α-fucose residues with α-fucosidase to forage on glycans provided by the mucus layer[33]. In addition, fucose on epithelial cells also serves as a receptor for pathogens, such as *Helicobacter pylori*[34], cholera toxin[35,36], norovirus[37,38], and rotavirus[39]. In this study, we focused on mucin fucosylation in the mechanism of intoxication by BoNT, the most potent known bacterial toxin. We showed that α1,2-fucosylation of mucin determined the entry route of BoNT in the small intestine (Fig. 5). The hyper-oral-toxic type of the BoNT complex (L-PTC/B$^{Okra}$) passed through the α1,2-fucosylated mucus barrier and entered the host through enterocytes within villous epithelium. By contrast, the non-hyper-oral-toxic toxin (L-PTC/A$^{62A}$) was trapped in the α1,2-fucosylated mucus layer. The mucus layer over the FAE is less thick than typical intestinal villous epithelium due to the absence of goblet cells[40]. Toxin that escapes from the mucus trap is absorbed by M cells within the FAE[8]. In addition, M cells express high levels of α1,2-linked fucose and GalNAc residues on their apical surface[41], which can serve as ligands for HA1/A$^{62A}$. Thus, non-hyper-oral-toxic toxins could selectively target M cells.

Humans with functional FUT2 are known as secretors. Approximately 20% of people worldwide are non-secretors who do not express histo-blood group antigens in mucus or other secretions in either the GI tract or saliva[29]. An individual's secretor status can affect their risk of

infections[32]. Secretors demonstrate a protective effect against infection with *H. pylori*[42], *Vibrio cholera*[43], enterotoxigenic *Escherichia coli*[44], norovirus[45], and rotavirus[46]. We explored whether and how the secretor status impacts human intestinal absorption of BoNT. Consistent with $Fut2$-null mice (Fig. 4b and Supplementary Fig. 10), HA/B$^{Okra}$ bound to human intestinal mucin from non-secretors more strongly than to that from secretors, although the difference was not significant ($p = 0.056$; Supplementary Fig. 11). These findings suggest that polymorphisms of the $FUT2$ gene could affect the susceptibility to oral BoNT intoxication. However, the epidemiological relationship between host secretor status and susceptibility to botulism remains unclear due to the lack of epidemiological data. Our findings might pave the way for clarifying susceptibilities to human botulism and could result in a deeper understanding of how genetic variations shape individual responses to infectious diseases. This knowledge could also facilitate the development of effective therapies against BoNT intoxication, as well as targeted oral delivery systems for drugs and antibodies[47,48].

## Methods

### Toxins and neurotoxin-associated proteins (NAPs)

Native BoNT and L-PTCs were produced by *C. botulinum* (A-62A[49], B-Okra, B-Osaka05[50]) and purified as previously described[51]. For biosafety considerations, active *bont* genes, in any form, were never expressed in *Escherichia coli*. Recombinant NAPs were produced by *E. coli* as previously described[5,19]. Briefly, NAP subcomponents (C-terminal FLAG-tagged HA1, N-terminal FLAG-tagged HA2, N-terminal *Strep*-tag II-tagged HA3, *Strep*-tag II-tagged HA3 in complex with His-tagged NTNHA) derived from A-62A or B-Okra were expressed by *E. coli* Rosetta 2 (DE3) and purified using appropriate affinity columns. HA, NAP, and L-PTC complexes were reconstituted and purified.

For immunofluorescence studies, native L-PTCs were labeled using an Alexa Fluor protein labeling kit (Thermo Fisher Scientific) according to the manufacturer's protocol. L-PTC/A$^{62A}$ and L-PTC/B$^{Okra}$ were labeled at 18.2- and 6.05-mole dye per mole protein, respectively. The labeled toxins were capable of inducing botulism in mice at comparable concentrations, albeit their i.p. toxicity was reduced approximately 10-fold compared to non-labeled toxins (Supplementary Fig. 12).

### Preparation of AfcA

His-tagged AfcA was prepared as previously described[26]. Briefly, *E. coli* Rosetta2 (DE3) cells harboring plasmid pET3a-AfcA were induced by an

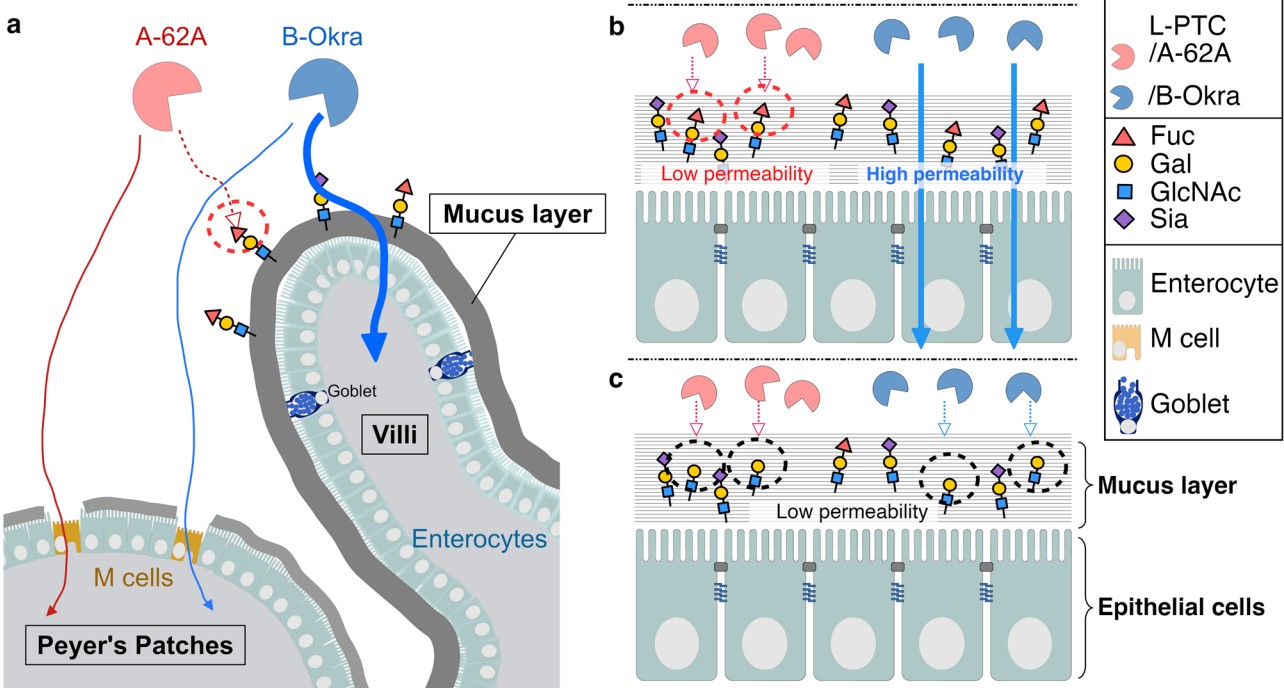

**Fig. 5 | Schematic model of the entry of botulinum neurotoxin complexes in the gut. a** Botulinum toxins from strain A-62A, a non-hyper-oral-toxic type, are captured by the intestinal mucin through interactions with α1,2-fucosylated glycans (red-dot circle), resulting in their entry via M cells of Peyer's patches. In contrast, toxins from strain B-Okra, a hyper-oral-toxic type, penetrate mucus layers and enter the host through enterocytes. **b**, **c** The levels of α1,2-fucosylation in intestinal mucins, which can be modulated by α1,2-fucosidase from gut bacteria, *Fut2* expression in mice, or the secretor status in humans, affect the ability of the hyper-oral-toxic toxins to penetrate the mucus barrier and exert oral toxicity. **b** In the gut of WT mice or secretor humans, the intestinal mucins are sufficiently glycosylated with fucose, preventing L-PTC/A⁶²ᴬ from penetrating through the mucus layers. **c** In Fut2-null mice, non-secretor humans, or after treatment with α1,2-fucosidase, the removal of terminal α1,2-fucosylation exposes the underlying galactose residues on the mucins, enabling both toxins to bind to the mucins (black-dot circle). Fuc fucose, Gal galactose, GlcNAc *N*-acetylglucosamine, Sia sialic acid.

auto-induction system. Recombinant AfcA was purified from the cell extract using a HisTrap HP column (Cytiva) and dialyzed against 10 mM Tris-HCl (pH 8.0).

## Mouse bioassay

Mice were maintained at ambient temperature under a 12-h light/dark cycle with *ad libitum* access to regular chow and water. Before administration of toxic L-PTCs, 7–10-week-old female BALB/c mice were maintained under fasting conditions for 4 h with free access to water. L-PTCs were diluted to various concentrations in 300 μL/mouse of bioassay buffer (10 mM sodium phosphate (pH 6.0), 0.1% gelatin). Mice were injected i.p. or gavaged i.g. with L-PTCs and observed every 12 h for up to 5 days for survival and signs of toxicity (ruffled fur, limb paralysis, general paralysis, pinched waist, labored breathing). Recombinant L-PTCs were administered at doses of 100 pg (intraperitoneal) or 200 ng (intragastric). In a mucin-depletion model, 7-week-old female BALB/c mice were gavaged with vehicle (sterile water) or 100 mg/mL NAC (Nacalai Tesque) in sterile water 1 h before toxin administration, and then gavaged with 50 ng of L-PTC/B^Okra or 500 ng of L-PTC/A^62A. Seven-week-old male and female *Fut2*-null (*Fut2*^−/−) mice were gavaged with 2 μg of L-PTC/A^62A and 3 ng of L-PTC/B^Okra. The *Fut2*^+/+ littermates were used as controls.

## Immunofluorescence

A ligated intestinal loop assay was performed as previously described[8]. Briefly, mouse ligated duodenal–jejunal loops were treated with 200 nM Alexa Fluor 488– or AF 568–labeled proteins. Two hours after treatment, the tissues were excised from the intestine and fixed with 4% PFA at room temperature for 30 min or with Carnoy's solution at room temperature for 1 h. The

PFA-fixed tissues were further permeabilized with 0.5% Triton X-100. The tissues were blocked with blocking buffer (2% BSA, PBS) for 1 h and incubated with primary antibodies (anti-GP2 mAb (MBL, 2F11-C3), anti-MUC2 pAb (Abcam, ab76774), followed by appropriate secondary antibodies conjugated with Alexa Fluor 405 (Abcam) or AF 488 (Thermo Fisher Scientific). Nuclei were probed with Hoechst 33342 (Dojindo). Slides were mounted with ProLong Antifade or ProLong Diamond Antifade (Thermo Fisher Scientific). Images were acquired by confocal microscopy using an IX71 microscope (Olympus) and a CSU21 or CSU-X1 scanner unit (Yokogawa), and analyzed using MetaMorph software (Molecular Devices).

## Isolation and purification of mucin

Mucus was harvested from the small intestine (duodenum and jejunum) by gentle scraping, and then resuspended in extraction buffer (6 M guanidine hydrochloride (GuHCl), 50 mM Tris-HCl (pH 8.0), 5 mM EDTA, 1× cOmplete™ Protease Inhibitor Cocktail (Roche)). After overnight rotation at 4 °C, the samples were centrifuged at 10,000 × *g* at 4 °C for 15 min, and then the supernatants were re-centrifuged at 100,000 × *g* at 4 °C for 1 h. The supernatants were diluted to 1.40 g/mL with CsCl and then subjected to density-gradient ultracentrifugation (Beckman MLS 50 rotor, 268,000 × *g*, 10 °C, 24 h). The samples were then subjected to a second density-gradient ultracentrifugation in CsCl/0.2 M GuHCl at a starting density of 1.42 g/mL. Mucin fractions were analyzed by dot blotting using biotinylated wheat germ agglutinin (WGA–biotin, MGC Woodchem) and Streptavidin-HRP (Jackson ImmunoResearch). Purified fractions were dialyzed against distilled water and stored at −80°C until use.

## Mucin ELISA

PGM (Merck, M1778), MIM, and human intestinal mucin were separately coated on 96-well ELISA plates (Iwaki). The plates were blocked with 1% BSA/PBS-T (pH 6.0) or Blocking One (Nacalai Tesque) and incubated with 10–200 nM L-PTCs, M-PTCs, or HAs at 37 °C for 1 h. After washing, the plates were incubated with primary antibodies (anti-BoNT/A, anti-BoNT/B, anti-FLAG (Merck, M2)), followed by appropriate secondary antibodies conjugated with horseradish peroxidase (Jackson ImmunoResearch). The plates were developed with ABTS (Merck). For a competition assay using carbohydrates, HAs were pre-incubated at 37 °C for 1 h with 10 mM carbohydrates (Gal, Wako, 071-0032; Lac, Wako, 128-00095; GalNAc, Nacalai, 00519-85; Neu5Ac (Sia), Nacalai, 00648-24; α2,3-sialyllactose (3SL), TCI, S0885; α2,6-sialyllactose (6SL), TCI, S0886; Fuc, Wako, 544-00111; 2FL, Carbosynth, OF06739; mannose (Man), Wako, 130-00872; Glc, Wako, 049-31165; GlcNAc, Nacalai, 00520-32). For a competition assay using lectins, the PGM-coated plates were pre-incubated at 37 °C for 2 h with 20 μg/mL biotinylated lectins (PNA; soybean agglutinin (SBA); DBA; concanavalin A (ConA); *Lens culinaris* agglutinin (LCA); *Lotus tetragonolobus* lectin (LTL); UEA-I; *Datura stramonium* agglutinin (DSA); WGA; *Macckia amurensis* mitogen (MAM); *Sambucus sieboldiana* agglutinin (SSA)) (all biotinylated lectins were purchased from MGC Woodchem) in lectin buffer (10 mM Tris-HCl (pH 7.4), 150 mM NaCl, 1 mM CaCl$_2$, 1 mM MgCl$_2$, 0.1 mM MnCl$_2$, 0.05% Tween 20). For α1,2-fucosylation removal, mucin-coated plates were pre-treated with 1 nM AfcA in 50 mM sodium phosphate (pH 6.0) at 30 °C for 24 h. For αGalNAcylation removal, PGM-coated plates were pre-treated with 20 units/well NAGA (NEB, P0734S) in 1× GlycoBuffer 1 and 1× BSA at 30 °C for 24 h.

## Mucin penetration assay

Transwell 24-well filters (0.4-μm pore; Corning) were coated with mucins at 50 °C for 2 h and then equilibrated with PBS (pH 6.0). L-PTCs at a concentration of 100 nM were carefully added to apical chambers. After incubation at 37 °C for 24 h, the permeate was harvested from basolateral chambers and analyzed by SDS-PAGE and immunoblotting with antibodies (anti–L-PTC/A, anti–L-PTC/B). Densitometry was quantified using FIJI ImageJ 1.53c. Control filters were coated with 2 μg/mL BSA.

## Crystallization and structure determination

Crystallization experiments were performed using the sitting drop vapor diffusion method at 22 °C on Cryschem plates (Hampton Research) and the droplets consisted of 3 μl of protein and 3 μl of reservoir solution. 5 mg/mL of *Strep*-tagII-tagged HA3/B in PBS (pH 7.4) was mixed with 1.6 mM 3SL or 6SL (Tokyo Chemical Industry). The crystals were grown in 14% polyethylene glycol (PEG) 3350 and 0.1 M Bis–Tris-HCl (pH 7.0), and then flash frozen using cryoprotectants containing 35% PEG 3350, 20 mM 3SL or 6SL, and 0.1 M Bis–Tris-HCl (pH 7.0). X-ray diffraction data were obtained at beamline BL44XU of SPring-8 and processed using HKL2000. The structures were solved by molecular replacement using an apo HA3/B structure (PDB ID: 3WIN[5]) as a search model using Molrep. Structure refinement was carried out using and Coot[52] and REFMAC5[53]. Initial refinement was performed using rigid-body refinement followed by restrained refinement with isotropic B-factors. After initial refinement, carbohydrate ligands were identified from the Fo-Fc maps at the 3.0σ level. In the final refinement stages, water molecules were automatically added to the models and manually curated. The data collection and refinement statistics are summarized in Supplementary Table 1. All molecular structure images were prepared using PyMOL (The PyMOL Molecular Graphics System, version 2.4.0; Schrödinger).

## Thermal shift assay

HA1/A and HA1/B were diluted to 2 μM in PBS (pH 7.4) and mixed with 0–200 mM lactose (FUJIFILM Wako) or α1,2-fucosyllactose (Biosynth)

in 20-μL total volume in the presence of 5× SYPRO Orange (Thermo Fisher Scientific). Fluorescence was detected from 25 to 95 °C in 0.5 °C/30-s steps using QuantStudio 3 (Thermo Fisher Scientific) with a ROX filter set. Melting curve and melting temperature ($T_m$) were analyzed using QuantStudio Design and Analysis Software v1.4.3. Dissociation constants ($K_D$) were estimated by plotting $T_m$ values versus the logarithm of ligand concentrations[54].

## Glycan microarray

Samples were labeled with Cy3 NHS ester monoreactive dye (Cytiva), and excess dye was removed with Sephadex G-25 desalting columns (Cytiva). Cy3-labeled proteins and mucins were applied a glycan microarray[55] (Supplementary Data 1). After overnight incubation at 20 °C, fluorescence signals were detected by an evanescent-field–activated fluorescence scanner (GP BioScience, GlycoStation reader 1200) and analyzed using an Array Pro Analyzer version 4.5 (Media Cybernetics).

## *FUT2* genotyping

Genomic DNA was extracted from specimens using a DNeasy Blood & Tissue Kit (QIAGEN). Genotyping (Se, secretor; se1, G428A; se2, A385T; se5, fusion/del) of *FUT2* was performed for 10 specimens using multiplex PCR[56]. For Se and se1, genomic PCR was performed using KOD One (TOYOBO) with specific primers: T5-F (TCTCCCAGCTAAC GTGTCCCG), T6-R (CAATCCCTGTCCACTCCGGCA), T7-F (TGGGCA TACTCAGCCCGTGT), and T8-R (CGGACGTACTCCCCCGGGAT). For Se, se2, and se5, genomic PCR was performed using EmeraldAmp (Takara Bio) with specific primers: T1-F (TGGGCATACTCAGCC CGTGT), NA1F (GGAGGAGGAATACCGCCACT), T2-R (CGGACGTAC TCCCCCGGGAT), and T5-F.

## Statistical and reproducibility

The glycan microarray and mouse bioassay with L-PTC/B$^{Osaka05}$ were evaluated at two different doses, whereas all other experiments were performed at least three times with similar results. All statistical testing was performed using RStudio (R version 4.1.2). Statistical significance was evaluated using the unpaired Student's *t*-test or log-rank test. Differences with $p < 0.05$ were considered statistically significant.

## Ethics statements

All animal experiments were approved by the animal experiment committee of Kanazawa University (AP-163710, AP-214252, AP-163708, AP-214251) and Research Institute for Microbial Diseases (RIMD) of Osaka University (H27-03-0), and were performed in accordance with the guidelines and regulations. Human samples of intestinal mucins and tissues were collected from autopsies at the Department of Forensic Medicine and Pathology, Graduate School of Medical Sciences, Kanazawa University. This study was approved by the Human Ethics Committee of Kanazawa University (2022-145).

## Reporting summary

Further information on research design is available in the Nature Portfolio Reporting Summary linked to this article.

# Data availability

Coordinates and structure factors of HA3/B in complex with 3SL and 6SL have been deposited in the Protein Data Bank (PDB) under accession codes 9UG5 and 9UG6, respectively. All experimental data are included in the article and the Supplementary Information. Source data are provided with this paper.

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

## Acknowledgements

We thank Yo Sugawara (National Institute of Infectious Diseases) for advice and discussion on HA mutants, and Keiko Hiemori for help with the microarray analysis. We also thank Chiyoko Aoki, Yuki Sano, Yuriko Tanaka, Sachiyo Akagi, Mayu Kitamura, Hitomi Kuraoka, and Yuki Konoshita for providing technical assistance and members of the Fujinaga laboratory for valuable discussion. *Fut2*-null mice were a general gift from Steven E. Domino (University of Michigan). We thank Eiki Yamashita and Atsushi Nakagawa (SPring-8 beamline BL44XU) for their help with the X-ray data collection. S.A. was supported by JSPS KAKENHI Grant Numbers 19K21257 and 23K14517. T.M. was supported by JSPS KAKENHI Grant Numbers 16K19123 and 18K07107. Y.F. was supported by JSPS KAKENHI Grant Number 24K02278. H.K. was supported in part by NIDDK grant P30 DK120515 and R01 DK051677, AMED grants 23ae0121040h0003, 223fa627003h0001, J22JK00035, J22JK00041, and J22JK00135 (cSIMVa) and the Chiba University-UC San Diego Center for Mucosal Immunology, Allergy, and Vaccines (cMAV).

## Author contributions

S.A., T.M., and Y.F. conceived the project and designed the experiments. S.A. and T.M. performed the majority of the experiments. S.A. performed competitive ELISA, isolation and purification of mucin, thermal shift assay, and *FUT2* genotyping. S.A. and K.K. performed structural analysis. C.M. prepared recombinant L-PTCs. S.K. and H.T. performed glycan and lectin microarray analysis. J.H. assisted with carbohydrate-binding analysis. T. Kohda provided L-PTC/B$^{Osaka05}$. T. Katayama provided the AfcA plasmid. Y.G. and H.K. provided *Fut2*-null mice. M.Z. provided human mucus samples. S.A. and Y.F. wrote the manuscript with editorial input from all the authors.

## Competing interests

The authors declare no competing interests.
