## [Transparent Peer Review file · Nature Communications]

Gut mucin fucosylation dictates the entry of botulinum toxin complexes

Corresponding Author: Professor Yukako Fujinaga

Version 0:

Reviewer comments:

Reviewer #1

(Remarks to the Author)

In consultation with the journal, I attempted to review this manuscript together with a junior scientist. Although we both belong to the research field of bacterial toxins, we found it extremely difficult to read and evaluate the manuscript.

We were both struck by the enormous range of experiments and data on which this manuscript is based. The work is certainly of high quality and significant. However, when reading the work, it is noticeable that the authors get lost in abbreviations and pay little attention to comprehensibility. There is no common thread and no logical structure to guide the reader through the experiments carried out. The figure legends and supplementary figures also do not help to gain a basic understanding of the work.

In particular, certain experiments lack controls, such as in Figure 1d, where no positive and negative controls for mucin permeability are shown. In Figure 3a, for example, it is not explained why some values are well above 100% and how the increase can be explained.

Overall, the manuscript appears to be a hodgepodge of different working groups that have contributed individual experimental data, which unfortunately could not be merged into a meaningful, readable manuscript.

We find it difficult to reject the manuscript because we have recognized the authors' efforts and the importance of the work. We would therefore like to suggest that the authors revise the manuscript completely and thoroughly, with a focus on comprehensibility and logical structure. Abbreviations should be reduced to a minimum as far as possible. We will then be available to review the revised version again.

Reviewer #2

(Remarks to the Author)

Reviewer #3

(Remarks to the Author)

Although being a rare disease in industrialized countries, foodborne-botulism is fatal and errors in industrial food processing can cause severe outbreaks with multiple patients requiring ventilation easily overwhelming hospitals' capacities. The causative agents are the botulinum neurotoxin complexes produced by *Clostridium botulinum* which are a diverse family of large 760 kDa protein complexes with 14 subunits. Although the 3D structures of the large progenitor toxin complexes (L-PTC) were characterized in great detail in the last decade, the actual mechanism of resorption in the intestine is only partially understood. In particular, the first steps of mucin penetration and epithelial transcytosis are still enigmatic. This manuscript addresses exactly these steps employing primary tissue and deciphering with multiple techniques the molecular details of mucin absorption and penetration as well as epithelial cell binding of L-PTC of serotypes A and B. They can demonstrate that the glycan specificity of the hemagglutinin HA1/HA33 subunit defines the distinctive behavior of L-PTC/A and L-PTC/B, the latter being ~80-fold more toxic. The presence of a 1,2-linked fucose in mucin mediates the absorption efficiency of L-PTC/A and B. They showed convincingly that the disruption of fucosylation in fucosyltransferase-2 (Fut2)-null mice impaired the oral toxicity of only L-PTC/B. In conclusion, this manuscript is a carefully conducted study significantly

advancing the molecular understanding of the first steps of food-borne botulism and even hypothesizing using preliminary data whether human non-secretors of FUT2 might be less susceptible to oral intoxication by L-PTC/B.

- Does the work support the conclusions and claims, or is additional evidence needed?

Yes, largely true. Improvements listed below.

- Are there any flaws in the data analysis, interpretation and conclusions? - Do these prohibit publication or require revision?
No

- Is the methodology sound? Does the work meet the expected standards in your field?

In principal yes, but:

- Fig. 1a, lines 74-77: The authors clearly show differences in oral toxicity of L-PTC/A-62A and L-PTC/B-okra but it is unclear how the stated 80-fold difference was calculated. Presented in vivo data should be sufficient to do the calculation which needs to be described in the M&M section.

- Fig. 1f: The success of NAC treatment vs vehicle is not obvious by MUC2 staining in case of L-PTC/A62A feeding. Authors should provide additional images.

- lines 112-115: The generation of recombinant chimeric L-PTCs is an elegant approach, however, the authors fail to provide any evidence that the assembly of the rL-PTC/BA which exhibited reduced oral toxicity is functional by providing an analytical SEC chromatogram and data of a pull down experiments demonstrating that BoNT/B, NTNHA/B and HA70-A are correctly interconnected at pH 6.0. Otherwise just mixing the components would yield the identical SDS-PAGE pattern as shown in Fig. 2d, but no 760 kDa L-PTC would have been formed thereby explaining reduced oral toxicity. Furthermore, a 4-fold higher dose of rL-PTC/B (200 ng) yields only 90% death whereas 50 ng native L-PTC/B yields 100% in 48 h. This could indicate non-quantitative assembly of the rL-PTC/B supporting above criticism and request for control data.

lines 124-126: According to Fig. 2g, only the mutation H281N/N282H in HA1 influences the PGM binding. Position 619 in HA3 is hardly relevant.

- lines 443-448: Varying dosages of L-PTC and rL-PTC in different experiments: In Fig. 1a, 1 µg L-PTC/A and 50 ng L-PTC/B yielded 80% and 100% death in 98 h and 48 h respectively. This data was nicely reproduced in Fig. 1g in the vehicle experiment. However, in Fig. 4c, 2 µg L-PTC/A (2-fold more) and 3 ng L-PTC/B (17-fold less) were used. Authors should explain why.

- lines 503-506, Fig. 1d and Fig. S2: The mucin penetration assay is very interesting and helps understanding the path of the L-PTC. However, the analytical method being used for detecting migrated L-PTC is far from quantitative. The immunoblotting using anti-L-PTC/A and anti-L-PTC/B antiserum is by best qualitative. Since the author established reliably working quantitative ELISAs for mucin binding assays I do not understand why these were not adopted for the mucin penetration assay. The data of Fig.1d/S2c is highly questionable.

- Is there enough detail provided in the methods for the work to be reproduced?

Following details need to be provided to allow reproduction of the work:

- Ethical statements: reference numbers of animal experiment approval and ethic committee statement should be listed, the latter being part of the source file.

lines 426-429: ref 37 is not available. In general, the use of recombinantly expressed L-PTC is a key part of this study. The expression, isolation, tag removal and especially L-PTC assembly process needs to be described in detail here. Especially the requirement of co-expression of HA3 with NTNHA is neither described here nor elsewhere in the references given.

lines 452-454: No details are provided regarding degree of labeling (DOL) of L-PTCs etc. and no statement is given regarding any functional impairment due to the labeling process. Excess labeling can greatly inactivate the biological activity and lead to false results. Authors should provide DOL and check remaining specific toxicity of label L-PTCs e.g. by MBA or alternative methods.

- Minor comments in the order of appearance

line 52: Hemagglutinin (HA) is understood as a single protein here and throughout the manuscript. Three of those HAs (HA1-HA3, HA33/HA17/HA70) form the 350 kDa 12-meric HA complex. Only HA1 and HA3 interact with carbohydrates. And only the HA3 trimeric complex and HA1 in the HA complex are mediating hemagglutination. Authors should use more precise language in the manuscript.

line 74: For completeness add LD50 data also for serotype F.

line 75: The term "hypervirulent" is misleading. Virulence relates to the organism's pathogenic potential. In botulism, the oral toxicity AND amount of expressed L-PTC yields virulence of *C. botulinum*.

line 77: During revision authors need to incorporate most recent structural and functional findings regarding serotype E PTCs (Gao et al. NSMB accepted Dec 18, 2024; Botulinum neurotoxins exploit host 1 digestive proteases to boost their oral toxicity via activating 2 OrfXs/P47).

line 205: To the best of my knowledge cholera toxin uses GM1 which is non-fucosylated as receptor

line 222-224: Fig. S10 describes binding of HA/A vs HA/B to HIM, not binding of L-PTC/B.

Fig. S3e: Is the data for 1 µg L-PTC/B and 1 µg L-PTC/A the same as in Fig. 1a? If so, state it.

Reviewer #4

(Remarks to the Author)

The manuscript by Amatsu et al. reveals an important factor for botulinum toxin toxicity, namely fucosylation of mucins. Except for the crystallographic analysis, the study is well executed and the conclusions solid. This reviewer suggests that the authors get professional help for the structural analysis – right now, the authors might just as well have included a manual docking study. In one of the structures, the R_{free} is twice as high as R_{work} (34% versus 17.5%) and thus clearly overrefined, in the other case still 8% higher. For the resolution, ranges should be given (and for the high-resolution range, all important data statistics; e.g., footnote b should be implemented). It does not make sense that the resolution of the data is lower than for the refined structure (although this is likely due to rounding; take care of precision used). CC1/2 is missing (R_{merge} is meaningless and given without resolution range). The number of outliers in the Ramachandran plot (7.4% and

3.7%) are very unlikely and (like the high B-factors) probably due to overrefinement and using an unreasonable weighting factor. Preferably, the word “redundancy” should be replaced with “multiplicity”, as the data are not redundant. R.m.s.d. = root mean square deviation from ideal values (add specification). PDB codes should be given, and validation sheets provided. In addition, a picture with electron density for the carbohydrate ligand should be shown and the refinement strategy described (i.e. ligand built at final step of refinement and density preferably shown before modeling it). It is also recommended that the authors use a better structure for their molecular replacement model. 3WIN has extremely poor statistics; a much better alternative would be 4OUJ (please do not forget to give the journal references for each of the PDB IDs throughout the manuscript; a comment on structure resolution or other quality indicators may also be helpful). In the Methods, I am also missing information about the protein solution used for crystallization (buffer, concentration), the volumes and ratio of the drops and reference to Table S1.

Methods, other: Please give the acid/base for adjusting the pH, e.g. Tris-HCl, as sometimes the counterions influence the experiment, as might be for the phosphate buffer.

Introduction: Is it correct to speak of toxin serotypes? Usually, it is bacteria (or viruses) that are serotyped depending on their properties. I would probably also be more cautious with the statement “are the most potent bacterial toxins” as more potent ones may exist but may not yet be known (better in discussion: “most potent known bacterial toxin”). Finally, I suggest to add some information about the hemagglutinins from the Results section, as this is central to this manuscript.

Results: To me, the argument about the “different intestinal routes” was not clear. This is probably due to my background, but others might also benefit from enhanced clarity of the argument.

On p.6/7, it was stated that a multiple sequence alignment classified HA into 3 sequence types of differing virulence. It is later explained why this makes sense, but at this point, this appears to be based on an assumption and lacks justification. The next subheader (Terminal ...) should be rephrased for discrimination, as it is only valid for type A.

The discussion is good, but some of the figures/legends can be improved:

- Fig. 1: b, describe the difference between the upper and lower panel. Add (red) and (green) after 568 and 488 labeling.
- Fig. 2: Replace “Bindings...were” with “Binding...was” (preferably also replace “and” with “or”. Add references for PDB IDs and add PDB ID for current structure (once well-refined). A supplementary figure with electron density for the carbohydrate ligand would help, as indicated above (alternatively, include density in Fig. 3e).
- Fig. 3: Consider adding a note in the legend about the specificity of UEA-1 and DBA. The color use makes it a bit difficult to see the methyl group of fucose, but I am not sure what can be done about this.
- Fig. S1: Not clear (to me) why the figure shows only uptake of B and not A (is it because of the concentration of the color in the middle? (might be helpful with a note)
- Fig. S3: remove second c HA3 (duplication confusing) or include “cont.”. Add information what is marked with the black arrowheads. What does “n = 5” mean here? Different strains tested?
- Fig. S4: PNA – correct legend, but Gal-GlcNAc is shown (needs correction). Check others. Add D/L (small caps) to all monosaccharide labels in a (with only fucose being an L-isomer).
- Fig. S6: There is nothing about simulations in the Methods. I assume the ligands were manually docked? Thus please revise the figure title. Also, “glycosylated Lac” sounds strange, since Lac itself is a sugar. Maybe better to say “Carbohydrate complexes of HA1”? Add references to PDB IDs. Remove alpha2,3/6-sialyllactose from legend, which is not shown.
- Fig. S7: done in triplicates? (mention)
- Fig. S8: mention that AfcA is an alpha-fucosidase (preferably also in the text). Repeat specificity of UEA-1 and PNA for easier reading. Generally, one also uses the terms “expression” and “mutation/mutants” for genes and “production” and “substitution/variants” for proteins.
- Fig. S9: What is meant with “Cont” and “Lots”? Purification of what? Reminding the specificity of WGA (and maybe UEA-1) in the text would also be helpful.
- Fig. S10: Was HIM introduced somewhere? I assume it means human intestinal mucin. Explain Se, se1, se2, se5. Explain red (A?)/blue (B?) coloring in c.
- Table S1 needs a complete overhaul and Table 2 is very small – split up?

Formalities:

- There should at least be a summary of the methods in the main article (right now this section is missing).
- The references in the Supplementary Information should either start from 1 or all references be included in the main manuscript, depending on Journal style. Also the references titles should be decapitalized.
- An abbreviation list would be very helpful.

Version 1:

Reviewer comments:

Reviewer #1

(Remarks to the Author)

Together with a junior colleague, I carefully reviewed the revised version of the manuscript. We found that the authors have substantially improved the readability of the text, making the manuscript clearer and more accessible. All of the points we previously raised have been addressed thoroughly and to our full satisfaction.

Furthermore, we noted that the authors have also responded exceptionally well to the concerns raised by the other referees, demonstrating a thorough and thoughtful approach to all feedback received.

Overall, we believe that the revised version is now ready for acceptance for publication.

Reviewer #2

(Remarks to the Author)

Reviewer #3

(Remarks to the Author)

The authors addressed all points raised in my first review largely sufficiently.

Two main points (term “hypervirulence” and describing generation and characterization of recombinant L-PTC/B) remain to become resolved, see below.

L82-3: not precise enough: Is the dose related to 150 kDa BoNT or the complex? For serotype F, authors list “M-PTC serotype F1: 2.4-3.6”. SI unit is missing and why here M-PTC instead of BoNT?

L84-5: “...the L-PTC from serotype B1-Okra (L-PTC/BOkra, termed hypervirulent) has at least 80-fold or 20-fold (Fig. 1a) higher oral toxicity than that from serotype A1-62A...”. Rewrite: “L-PTC/BOkra has 20-80-fold higher oral toxicity than that...”. Again, virulence is defined as the strength of pathogenicity of a virus or bacteria. *C. botulinum* is pathogenic due to the expression of BoNT. The potency/specific toxicity of the BoNT in conjunction with its expression levels define the virulence of the *C. botulinum* strain. I strongly recommend to replace hypervirulence by increased oral potency.

Apart from the term, what is your definition for “hypervirulence”? Do you set an absolute minimum oral toxicity in µg/kg bodyweight of 150 kDa BoNT and everything above is “hypervirulent”? And is then M-PTC/F “hypovirulent”?

L121: Change “The M-PTC from neither toxin bound to PGM” to “...from neither serotype...”

L125-7: Cite “A nut-and-bolt assembly of the bimodular large progenitor botulinum neurotoxin complex” Lam et al. accepted in *Science Advances* which showed inter-serotype combination of NTNHA and HA70.

L131: I checked the Toxins manuscript describing generation and characterization of recombinant L-PTC/B. The authors need to make sure that the Toxins manuscript will be published prior to this manuscript otherwise the Material & Method section of this manuscript is insufficient by lacking the generation and characterization of recombinant L-PTC/B which is an integral part of the study.

L233: Change “pathogenicity of BoNT” to “mechanism of intoxication by BoNT”

Reviewer #4

(Remarks to the Author)

Clearly, the manuscript is much improved in all respects and reports important results. Still there are aspects that can be further improved.

With respect to the crystallographic analysis, I am basing my judgement on Table 2, the text in the manuscript and the validation reports as the PDB entries are not yet released. Both structures are significantly improved. The low-resolution structure (PDB ID: 9UG5) still has many weak aspects, among others the very high Rmerge (10.5%) and B-factors. It is quite possible that there is not much more to achieve with this data set, although the authors might like to double check if the space group is potentially only pseudo-orthorhombic (try e.g. P21). Good is the electron density fit of the carbohydrate ligand. However, it is very surprising that the authors chose to build water molecules into 3 Å density. This is not warranted until appr. 2.5 Å resolution. The authors might like to double check if these are ions. – The 2.2 Å structure (PDB ID: 9UG6), due to its higher resolution, is of better overall quality, however, here the electron density for the carbohydrate ligand is much less convincing, with large patches of negative difference density indicating errors. I wonder at which state the authors built in the ligand (as this is not commented)? Was it at the very end of the refinement or early on and thus not unbiased? The authors might like to remove the ligand, run simulated annealing and then add back the ligand into unbiased density or use Polder maps. For this entry, I observed a discrepancy between the Rmerge in Table 2 (4.7%) and in the validation report (6%). This should be clarified. The difference between R and Rfree (7.4%) is very high and it should be possible to improve this significantly given the data quality. (While they are at it, the authors might also like to improve their previous structure,

3WIN, which they used for molecular replacement). – Apart from this, I suggest the following improvements of Table 2: Give 3-4 digits for the wavelength, but only one digit for unit cell parameters (except crystallographic angle), completeness, Ramachandran distribution and B-factors. Add unit (Å²) for B-factors. The Methods section could also be further improved: Report the refinement strategy (e.g. when were ligands and water molecules added and according to which criteria); clarify what exactly the cryo-protectant contained (also buffer? And I assume the respective sialyllactose ligands 3SL and 6SL); and remove the redundancy at the start of the section (e.g., shorten to “...(Tokyo Chemical Industry). Crystallization was performed using Cryschem plates...and the droplets consisted of 3 ul protein + ligand and 3 ul reservoir solution containing xxx.”

Overall, I would like to commend the authors for improving the readability of the manuscript. Still, with my background, the first part of the manuscript was challenging to read. For example, I would benefit from additional details in the legend to Figure 1, explaining early on that vehicle consists of sterile water and what GP2 is. Given that fucosylation is so important, I also recommend to add more references in the Discussion (p. 12, lines 230-232). For example, many people do not even know about the importance of fucosylated cholera toxin receptors (as reported by Wands et al., Cervin et al and many papers from the Krenzel lab). By the way, secretors are also protected from cholera (line 248). Finally, to drive the main points home, the authors may consider adding as Figure 5 a cartoon drawing of the main findings.

Minor points for correction:

- Introduction:

o Line 68: Is there a consequence of combination with OrfX2 protein? (extension of the sentence could be helpful as it is difficult to read).

o HA: I was mainly aware of HA in the influenza virus. Add an additional explanatory sentence?

o Line 85: “at least 80-fold or 20-fold” sounds strange – rephrase to “more than 20-fold and give reference and call figure thereafter?”

- Line 94: How was it shown that L-PTC/Bokra underwent endocytosis? (not clear to me, but probably obvious to cell biologists)

- Line 130: Sentence strange (equivalent?)

- Line 167: Should be Fig. 3 (not S5)

- Methods: add where the sugars and lectins were purchased from (only done for some of the compounds)

- List abbreviations alphabetically (easier to find)

- Decapitalize references

- Figures:

o Fig. 1: Lines 578-580 (d should also apply to f and h (log-rank test) to a. Add specification of vehicle and GP2.

o Fig. 2: Consider referring to Fig. S4 in f.

o Fig. 3: It could be helpful to refer to Fig. S5 in the legend. Line 616: It should be Fig. S5 (not 5) and (43) should be deleted. Line 620: Not completely clear – refer to crystal structures 4LO2 and 4OUJ and docked complex with fucose added (see Fig. S7).

o Fig. 4c: Consider Making symbol for Fut2^{-/-} half red and half blue.

o Fig. S2: Line 678 (T ->t). Line 682 (B -> b). And: I am not quite sure if it is okay to normalize to BSA-coated wells, giving >100% in permeability in c?

o Fig. S5: Much appreciated! – Can be further improved formally, by using convention of small caps for D/L. Line 708: Legend refers to Gal-beta 1,3 GalNAc specificity of PNA, but figure shows Gal-beta 1,3 GlcNAc. SBA: GalNAc and Gal-specific. ConA: Man- and Glc-specific. SSA: misleading, as it is not GalNAc-specific (add Sia). Fig. S9 Refer to SNFG symbol nomenclature (<https://www.ncbi.nlm.nih.gov/glycans/snfg.html>).

o Fig. S9: Line 741: Maybe better write “Recombinant wild-type a1,2 fucosidase AfcA (WT) to avoid the impression that these are two difference versions of the enzyme?”

o Fig. S10: revise nul -> null

Version 2:

Reviewer comments:

Reviewer #4

(Remarks to the Author)

I am pleased with the authors' revision and only have three small comments left:

a) on p. 13 l. 250 (Discussion), it should be [infection with] *Vibrio cholerae* - not Cholera

b) typo in the legend of Fig. S5 (p. 42, l. 753): *Samubucus* -> *Sambucus* (remove u)

c) I recommend that the beautiful Fig. S13 is moved to the main manuscript.

In addition, as encouraged by the journal, I shared the manuscript with a junior researcher, who had the following comments:

The manuscript presents very interesting results about the role of mucin fucosylation in BoNT intoxication. The work is supported by many different experiments, including cell biology, binding assays, toxicity studies, and even some structural biology. It was nice to read. I particularly liked supplementary figure 13 showing the model of how the toxins pass through the mucin and enter the host. I think this figure should be moved to the main text, as it summarizes the findings quite well. I also liked that the authors included some structures of the toxins bound to sugars, but I would have liked to see more discussion about why HA1/A62A can accommodate the fucosylated sugar better than HA1/Bokra.

I have some additional suggestions that could further improve the paper:

-In the abstract (line 50), the authors could specify that B-Okra refers to a serotype.

-It would be nice to add a figure (early in the text) showing the general architecture of the L-PTC complexes (similar to the one shown in Fig2.d for the toxin chimeras). They are described at the beginning of the introduction, but a figure can help the readers visualize the individual components and how they are arranged, making the paper easier to follow from the start.

-The introduction ends in the knowledge gap, but it could be a good idea to add a couple of sentences about the research question and the main findings. This is already mentioned in the abstract, but it could be stated again in this section.

-In the methods section 'Toxins and neurotoxin-associated proteins (NAPs)', maybe specify that the NAP components were produced separately in *E. coli* (and not co-expressed).

-In the results, it was not made clear which structures were obtained in this work. At least in Figure 2.f, line 627, add '(this work)' after the new structure is mentioned.

Minor points for correction:

-Line 114: 'mutations' is repeated twice.

-Line 145: 'WT vs. 619 mutants' sounds strange. Write either 'residue 619 mutants', or Arg/Lys 619 mutants to make it clearer.

-Line 322: write full name of reagent: 1x cOmplete™ Protease Inhibitor Cocktail

-Line 644: make the letter 'b' in the figure legend bold.

-Lines 626, 627 and 629: α 2-6-sialyllactose is abbreviated as 'SL6', but in the rest of the text it is '6SL'. Change to 6SL for consistency.

Reviewer #1/2 (Remarks to the Author):

In consultation with the journal, I attempted to review this manuscript together with a junior scientist. Although we both belong to the research field of bacterial toxins, we found it extremely difficult to read and evaluate the manuscript. We were both struck by the enormous range of experiments and data on which this manuscript is based. The work is certainly of high quality and significant. However, when reading the work, it is noticeable that the authors get lost in abbreviations and pay little attention to comprehensibility. There is no common thread and no logical structure to guide the reader through the experiments carried out. The figure legends and supplementary figures also do not help to gain a basic understanding of the work.

We added the sentences in the main manuscript to help readers to understand results and our objectives (P6L97-101, P7L128-P8L131, P9L163-165, P10L178-179, P10L190, P11L205-206).

In particular, certain experiments lack controls, such as in Figure 1d, where no positive and negative controls for mucin permeability are shown.

The results in Fig 1d are "relative" mucin permeability. We utilized control (BSA)- or mucin-coated culture inserts and the permeabilities of toxins through the mucin were normalized to those through BSA. For clarity, we added the data of BSA-coated wells (control) to Fig. S2 (P36). The band intensities of intact BoNT and HA1 from PGM- and MIM-coated wells were normalized to those from BSA-coated wells.

In Figure 3a, for example, it is not explained why some values are well above 100% and how the increase can be explained.

The values above 100% do not necessarily reflect specific enhancement effects. While the binding of B-Okra to the mucin was generally low and showed some variability, these carbohydrates did not exhibit an inhibitory effect. Therefore, to ensure a representative depiction of the inhibition profile, we additionally performed the inhibition ELISA under identical conditions and substituted the results with a representative dataset (Fig. 3a) (P31).

Overall, the manuscript appears to be a hodgepodge of different working groups that have contributed individual experimental data, which unfortunately could not be merged into a meaningful, readable manuscript. We find it difficult to reject the

manuscript because we have recognized the authors' efforts and the importance of the work. We would therefore like to suggest that the authors revise the manuscript completely and thoroughly, with a focus on comprehensibility and logical structure.

To enhance readability and clarify the comprehensive connections within our work, we added explanatory sentences to facilitate a clearer understanding of the results as described above. We also spelled out abbreviations as much as possible for smoother reading.

Abbreviations should be reduced to a minimum as far as possible.

For easier reading, we spelled out the abbreviations as well as possible and added an abbreviation list (P20L406-P21L418).

We will then be available to review the revised version again.

Reviewer #3 (Remarks to the Author):

- Does the work support the conclusions and claims, or is additional evidence needed?

Yes, largely true. Improvements listed below.

- Are there any flaws in the data analysis, interpretation and conclusions? - Do these prohibit publication or require revision?

No

- Is the methodology sound? Does the work meet the expected standards in your field?

In principal yes, but:

- Fig. 1a, lines 74-77: The authors clearly show differences in oral toxicity of L-PTC/A-62A and L-PTC/B-okra but it is unclear how the stated 80-fold difference was calculated. Presented in vivo data should be sufficient to do the calculation which needs to be described in the M&M section.

According to the previous study (ref. 1; Sakaguchi. 1982), L-PTC/A-62A showed 80-fold higher oral toxicity than L-PTC/B-Okra. Our results shown in Fig. 1a also confirmed this higher oral toxicity (at least 20-fold higher). For clarity, we revised the sentence (P6L85).

- Fig. 1f: The success of NAC treatment vs vehicle is not obvious by MUC2 staining in case of L-PTC/A62A feeding. Authors should provide additional images.

To confirm the effect of NAC, we quantified the amount of mucin in samples of mouse small intestine. We collected the mucin from vehicle- or NAC-treated mice small intestines and performed ELISA using WGA. Our results demonstrated that NAC treatment resulted in a decrease in mucin levels. We added the new result (Fig. 1f) (P27) and sentence (P28L573-575).

- lines 112-115: The generation of recombinant chimeric L-PTCs is an elegant approach, however, the authors fail to provide any evidence that the assembly of the rL-PTC/BA which exhibited reduced oral toxicity is functional by providing an analytical SEC chromatogram and data of a pull down experiments demonstrating that BoNT/B, NTNHA/B and HA70-A are correctly interconnected at pH 6.0. Otherwise just mixing the components would yield the identical SDS-PAGE pattern as shown in Fig. 2d, but no 760 kDa L-PTC would have been formed thereby explaining reduced oral toxicity.

We administered the recombinant toxins that were reconstituted and purified. We have confirmed that these complexes were potent. There are three evidences; First, our method yielded the chimeric toxin complexes with high purity (Fig. 2d). Second, the i.p. toxicities were comparable between chimeric toxins (Fig.2e), indicating these toxins contained equal amount of BoNT/B. Third, HA in these complexes showed the comparable barrier-disrupting activity (caused by the E-cadherin-binding activity of HA; ref. 6-8, 10,11) and each complex bound to the mucin as each HA did (provisionally indicated as ref.18). We are planning to submit a manuscript on reconstituted L-PTC entitled "**Hypervirulent Botulinum Toxin Complex Exerts Oral Toxicity by Disrupting the Intestinal Epithelial Barrier**" (ref.18), and attached this manuscript. Please check this.

Furthermore, a 4-fold higher dose of rL-PTC/B (200 ng) yields only 90% death whereas 50 ng native L-PTC/B yields 100% in 48 h. This could indicate non-quantitative assembly of the rL-PTC/B supporting above criticism and request for control data.

As the reviewer mentioned, the i.g. and i.p. toxicity were different between native and recombinant L-PTC, but there was no difference between rL-PTC/BB and rL-PTC/BA in i.p. toxicity. We are confident that rL-PTCs were quantitatively assembled as shown in ref.18 Fig.1 and 2. We assumed that the difference is due to the assembly efficiency of BoNT and NAPs because the efficiency is not 100%. As described above, we confirmed that these recombinant toxin complexes contained the comparable activity of BoNT shown in Fig. 2e (i.p. toxicity). We added the sentence for notifying the difference between native and recombinant L-PTCs (P7L128-P8L131).

lines 124-126: According to Fig. 2g, only the mutation H281N/N282H in HA1 influences the PGM binding. Position 619 in HA3 is hardly relevant.

We agree that the effect of 619 in HA3 is much smaller than that of 281/282 in HA1. So, we described "primarily in HA1" (P8L143) and focused on HA1 thereafter. Besides, we cannot ignore the effect of 619 in HA3 because the R528A (sialic acid-binding deficient) mutant significantly decreased the HA binding to PGM (Fig. 2g). In addition, the mutations, R619K in HA3/A and K619R in HA3/B, also made small but significant changes in the binding ability (vs. WT; $p = 0.057$ (A-62A), 0.037 (B-Okra)). This explanation has also been incorporated into the main manuscript (P8L144-146).

- lines 443-448: Varying dosages of L-PTC and rL-PTC in different experiments: In

Fig. 1a, 1 µg L-PTC/A and 50 ng L-PTC/B yielded 80% and 100% death in 98 h and 48 h respectively. This data was nicely reproduced in Fig. 1g in the vehicle experiment. However, in Fig. 4c, 2 µg L-PTC/A (2-fold more) and 3 ng L-PTC/B (17-fold less) were used. Authors should explain why.

We usually administer mice with the toxins at the variable dosage that definitely kills almost all normal mice within 5 days-post gavage: 500–2000 ng for A-62A, 5-50 ng for B-Okra. It is because we used the native toxins purified from *C. botulinum* as previously described in ref. 43. The toxins were naturally activated by *C. botulinum* (ref. 1), although the activation extent is dependent on the purification batch. We confirmed the toxic activity of toxins from each purification batch and subsequently used in mouse bioassay. And, we used the same batch of toxins for the same experiment to compare the toxicity.

- lines 503-506, Fig. 1d and Fig. S2: The mucin penetration assay is very interesting and helps understanding the path of the L-PTC. However, the analytical method being used for detecting migrated L-PTC is far from quantitative. The immunoblotting using anti-L-PTC/A and anti-L-PTC/B antiserum is by best qualitative. Since the author established reliably working quantitative ELISAs for mucin binding assays I do not understand why these were not adopted for the mucin penetration assay. The data of Fig.1d/S2c is highly questionable.

As the reviewer pointed out, ELISA should provide a more quantitative assessment. We performed mucin ELISA using permeabilized samples, but all samples were under the detection limits. On the other hand, Western blotting is also effective for evaluating the relative amounts of the protein. As shown in Fig. 1d/S2, all proteins were loaded on the same membrane and detected using anti-L-PTC antiserum, allowing to compare the relative mucin penetration. Furthermore, we can confirm the migrated molecular compositions (BoNT, NTNHA, and HA) by Western blotting using anti-L-PTC.

- Is there enough detail provided in the methods for the work to be reproduced?

Following details need to be provided to allow reproduction of the work:

- Ethical statements: reference numbers of animal experiment approval and ethic committee statement should be listed, the latter being part of the source file.

We added these numbers (P14L265, 266, 270).

lines 426-429: ref 37 is not available. In general, the use of recombinantly expressed L-PTC is a key part of this study. The expression, isolation, tag removal

and especially L-PTC assembly process needs to be described in detail here. Especially the requirement of co-expression of HA3 with NTNHA is neither described here nor elsewhere in the references given.

We apologize for our clumsiness at the first submission. We are planning to submit a manuscript on reconstituted L-PTC entitled "**Hypervirulent Botulinum Toxin Complex Exerts Oral Toxicity by Disrupting the Intestinal Epithelial Barrier**" (ref.18), and attached this manuscript.

lines 452-454: No details are provided regarding degree of labeling (DOL) of L-PTCs etc. and no statement is given regarding any functional impairment due to the labeling process. Excess labeling can greatly inactivate the biological activity and lead to false results. Authors should provide DOL and check remaining specific toxicity of label L-PTCs e.g. by MBA or alternative methods.

We added the information on the degree of labeling (P14L279-P15L284). We also evaluated the biological activity of the labeled toxins and added the results in the supplementary figure (Fig. S12) (P49). The labeling reduced the i.p. toxicity 10-fold, but we confirmed the labeled A-62A and B-Okra toxins could induce botulism in mice at comparable concentrations.

- Minor comments in the order of appearance

line 52: Hemagglutinin (HA) is understood as a single protein here and throughout the manuscript. Three of those HAs (HA1-HA3, HA33/HA17/HA70) form the 350 kDa 12-meric HA complex. Only HA1 and HA3 interact with carbohydrates. And only the HA3 trimeric complex and HA1 in the HA complex are mediating hemagglutination. Authors should use more precise language in the manuscript.

We described HA as a protein complex overall the manuscript. For clarity, we added the sentence "*hemagglutinin (HA) complex, which is one of the L-PTC components.*" in abstract section and "*HA is a large protein complex which comprises three subcomponents: HA1, HA2, and HA3 (also known as HA33, HA17, and HA70, respectively)*" in introduction section (P5L71-73).

line 74: For completeness add LD50 data also for serotype F.

We added it (P5L83).

line 75: The term "hypervirulent" is misleading. Virulence relates to the organism's

pathogenic potential. In botulism, the oral toxicity AND amount of expressed L-PTC yields virulence of *C. botulinum*.

As the reviewer pointed out—"In botulism, the oral toxicity AND amount of expressed L-PTC yields virulence of *C. botulinum*"—this study focuses on "oral toxicity". We used the term hypervirulent "toxin", not "strain", to emphasize this point.

Importantly, there is no substantial difference in toxin production between strains A-62A and B-Okra (Sugii, et al. *Infect Immun* (1975) doi: 10.1128/iai.12.6.1262-1270.1975; Kozaki, et al. *Infect Immun* (1974) doi: 10.1128/iai.10.4.750-756.1974) and their purified toxins show similar i.p. toxicities. These data suggest that the oral toxicity of the toxin is a key determinant of virulence in botulism. However, certain strains, e.g. Hall A-hyper (Bradshaw, et al. *Anaerobe* (2004) doi: 10.1016/j.anaerobe.2004.07.001), are known to produce higher levels of toxin. Moreover, as the reviewer pointed out, the term "hypervirulent" is generally used to describe pathogens (bacteria and viruses) rather than individual toxins. If the current terminology is deemed inappropriate, we are ready to revise "hypervirulent-type toxin" to another term, e.g. "high oral toxicity-type toxin", including an associated manuscript (ref. 18).

line 77: During revision authors need to incorporate most recent structural and functional findings regarding serotype E PTCs (Gao et al. NSMB accepted Dec 18, 2024; Botulinum neurotoxins exploit host 1 digestive proteases to boost their oral toxicity via activating 2 OrfXs/P47).

We added descriptions about OrfX (P5L67-68, L75-76).

line 205: To the best of my knowledge cholera toxin uses GM1 which is non-fucosylated as receptor

GM1 is the most known receptor for cholera toxin. We did not dismiss GM1. GM1 is a strong ligand, but poorly expressed in the intestinal epithelial cells. Recently, Wang, et al. have reported that fucosylated HBGA, which is abundant in intestinal epithelial cells, can serve as the host receptor in the intestine. Please take a look at the paper (<https://doi.org/10.1021/acsinfecdis.7b00085>).

line 222-224: Fig. S10 describes binding of HA/A vs HA/B to HIM, not binding of L-PTC/B.

We revised it (P13L250).

Fig. S3e: Is the data for 1 µg L-PTC/B and 1 µg L-PTC/A the same as in Fig. 1a? If so, state it.

These are the same figures. We added the sentence " *The survival data of mice administered with L-PTC/BOkra and L-PTC/A62A are from the same experiment as Fig. 1a.* " in the legend of Fig. S3 (P39L693-695).

Reviewer #4 (Remarks to the Author):

The manuscript by Amatsu et al. reveals an important factor for botulinum toxin toxicity, namely fucosylation of mucins. Except for the crystallographic analysis, the study is well executed and the conclusions solid. This reviewer suggests that the authors get professional help for the structural analysis – right now, the authors might just as well have included a manual docking study. In one of the structures, the R_{free} is twice as high as R_{work} (34% versus 17.5%) and thus clearly overrefined, in the other case still 8% higher. For the resolution, ranges should be given (and for the high-resolution range, all important data statistics; e.g., footnote b should be implemented). It does not make sense that the resolution of the data is lower than for the refined structure (although this is likely due to rounding; take care of precision used).

CC1/2 is missing (R_{merge} is meaningless and given without resolution range).

Our data was collected 10 years ago and analyzed by HKL2000, which lacked CC1/2 values. Alternatively, we added the Mean I/σ(I) values to Table S1 (P50).

The number of outliers in the Ramachandran plot (7.4% and 3.7%) are very unlikely and (like the high B-factors) probably due to overrefinement and using an unreasonable weighting factor.

We have refined the structure and decreased the number of outliers in the Ramachandran plot (0.66 and 0.46).

Preferably, the word “redundancy” should be replaced with “multiplicity”, as the data are not redundant. R.m.s.d. = root mean square deviation from ideal values (add specification).

We revised it (Table S1) (P50).

PDB codes should be given, and validation sheets provided.

We deposited the structure to PDB (9UG5, 9UG6), and attached validation sheets.

In addition, a picture with electron density for the carbohydrate ligand should be shown and the refinement strategy described (i.e. ligand built at final step of refinement and density preferably shown before modeling it).

We created a new figure (Fig. S4) drawing the omit map (P40).

It is also recommended that the authors use a better structure for their molecular replacement model. 3WIN has extremely poor statistics; a much better alternative would be 4OUJ (please do not forget to give the journal references for each of the PDB IDs throughout the manuscript; a comment on structure resolution or other quality indicators may also be helpful).

We appreciate the reviewer's advice, but 4OUJ is a structure of HA1/B, not HA3/B.

In the Methods, I am also missing information about the protein solution used for crystallization (buffer, concentration), the volumes and ratio of the drops and reference to Table S1.

We added more information about crystallization in the method section (P18L360-363).

Methods, other: Please give the acid/base for adjusting the pH, e.g. Tris-HCl, as sometimes the counterions influence the experiment, as might be for the phosphate buffer.

We added the acid/base information (P15L289, P16L320, P17L345, P18L360, P18L364).

Introduction: Is it correct to speak of toxin serotypes? Usually, it is bacteria (or viruses) that are serotyped depending on their properties.

We speak of toxin serotypes. In general, as the reviewer mentioned, the term "serotype" is used for bacteria, but in botulism botulinum toxins are classified by serotypes with neutralizing antibodies.

I would probably also be more cautious with the statement "are the most potent bacterial toxins" as more potent ones may exist but may not yet be known (better in discussion: "most potent known bacterial toxin").

We revised it (P4L49).

Finally, I suggest to add some information about the hemagglutinins from the Results section, as this is central to this manuscript.

We added the information about HA in the introduction section (P5L71-73).

Results: To me, the argument about the "different intestinal routes" was not clear.

This is probably due to my background, but others might also benefit from enhanced clarity of the argument.

For clarity, we added the sentence in the results section (P6L97-101).

On p.6/7, it was stated that a multiple sequence alignment classified HA into 3 sequence types of differing virulence. It is later explained why this makes sense, but at this point, this appears to be based on an assumption and lacks justification.

We revised the manuscript: First, we described the sequence types of HA (P8L151-P9L153). Then, we discuss oral toxicities by referring to the results of mouse bioassay with L-PTC/A-62A, /B-Okra, and /B-Osaka05 (P9L153-157).

The next subheader (Terminal ...) should be rephrased for discrimination, as it is only valid for type A.

Terminal fucosylation of the intestinal mucin is critical for not only type A-62A but also type B-Okra. L-PTC/A-62A was trapped in the mucin by binding to the terminal fucosylation. By contrast, L-PTC/B-Okra, which is not capable of binding to terminal fucosylation, is easy to pass thorough the mucin. If the terminal fucosylation is removed, L-PTC/B-Okra is also trapped in the mucin by binding to terminal galactose.

The discussion is good, but some of the figures/legends can be improved:

- Fig. 1: b, describe the difference between the upper and lower panel. Add (red) and (green) after 568 and 488 labeling.

We have already described the difference between the upper and lower panels in the legend of Fig. 1, but we revised for clarity as follows *"Scale bars, 50 μm (low magnification) in upper panels of b and g; 20 μm (high magnification) in lower panels of b and c."* (P27L566-568).

We added the color name of the protein (P27L563-564).

- Fig. 2: Replace "Bindings...were" with "Binding...was" (preferably also replace "and" with "or").

We rephrased it (P29L584).

We also replace other "bindings" with "binding" (P28L575, P30L595, P31L605, P32L628, P33L634, L636, L637).

Add references for PDB IDs and add PDB ID for current structure (once well-refined).

We added the references to PDB ID (P18L368, P30L592, P44L729).

A supplementary figure with electron density for the carbohydrate ligand would help, as indicated above (alternatively, include density in Fig. 3e).

We created a new figure (Fig. S4) drawing the omit map (P40).

- Fig. 3: Consider adding a note in the legend about the specificity of UEA-1 and DBA. The color use makes it a bit difficult to see the methyl group of fucose, but I am not sure what can be done about this.

We added the sentence about specificity of lectins in the legend (P31L610-620).

- Fig. S1: Not clear (to me) why the figure shows only uptake of B and not A (is it because of the concentration of the color in the middle? (might be helpful with a note)

L-PTC/B-Okra, but not A-62A, was endocytosed into enterocytes in the villous epithelium, as it appeared as puncta (white arrowhead in Fig. S1). For clarity, we added the sentence in the legend of Fig. S1 (P35L669-670).

- Fig. S3: remove second c HA3 (duplication confusing) or include "cont.".

We revised it.

Add information what is marked with the black arrowheads.

The information of the black arrowhead has already been described in the legend of Fig. S3 (P38L687-688).

What does "n = 5" mean here? Different strains tested?

As described in the legend of Fig. S3, BALB/c mice (n = 5) were challenged with the toxins (P39L691).

- Fig. S4: PNA – correct legend, but Gal-GlcNAc is shown (needs correction).

Check others.

We intended to describe the specificity of lectin; PNA recognizes Gal-beta1,3-GlcNAc. For clarity, we added the "-specific lectin" (P41L708-P42L717).

Add D/L (small caps) to all monosaccharide labels in a (with only fucose being an L-isomer).

We added the D/L information (P41L703-704).

- Fig. S6: There is nothing about simulations in the Methods. I assume the ligands were manually docked? Thus please revise the figure title.

We manually docked the ligands. We changed the figure title (P44L727).

Also, “glycosylated Lac” sounds strange, since Lac itself is a sugar. Maybe better to say “Carbohydrates of HA1”?

We revised it; from “glycosylated Lac” to “carbohydrates” (P44L729).

Add references to PDB IDs.

We added the references to PDB ID (P44L729).

Remove alpha2,3/6-lactose from legend, which is not shown.

We removed it (P44L729).

- Fig. S7: done in triplicates? (mention)

The thermal shift assay was done in quadruplicate. We mentioned it in the legend of Fig. 3 and S7 (P32L624, P45L736).

- Fig. S8: mention that AfcA is an alpha-fucosidase (preferably also in the text).

We revised it (P32L625, P46L741).

Repeat specificity of UEA-1 and PNA for easier reading.

For easier reading, we added the specificity of lectins with the lectin name as long as possible overall manuscript.

Generally, one also uses the terms “expression” and “mutation/mutants” for genes and “production” and “substitution/variants” for proteins.

We changed the word "expression and purification of" to "preparation of" (P15L285, P46L741).

- Fig. S9: What is meant with “Cont” and “Lots”? Purification of what?

We revised these term; from "Cont" to "WT", "Lots" to "from three different mucin purification lots" (P46L749-751).

Reminding the specificity of WGA (and maybe UEA-1) in the text would also be helpful.

We remind of the specificity of lectin in the legend of Fig S9 (P46L752-753).

- Fig. S10: Was HIM introduced somewhere? I assume it means human intestinal mucin.

We spelled out HIM to human intestinal mucin (HIM) (P48L760).

Explain Se, se1, se2, se5.

We added the explanation of genotypes (Se, se1, se2, se5) (P48L761-762).

Explain red (A?)/blue (B?) coloring in c.

We added the explanation of bar charts (P48L764-765).

- Table S1 needs a complete overhaul

We have refined the structure and updated the Table S1 (P50).

and Table 2 is very small – split up?

We split up the Table S2 (P51, P52).

Formalities:

- There should at least be a summary of the methods in the main article (right now this section is missing).

We moved the method section from supplemental information to main manuscript.

- The references in the Supplementary Information should either start from 1 or all references be included in the main manuscript, depending on Journal style. Also the references titles should be decapitalized.

We revised it.

- An abbreviation list would be very helpful.

We added the abbreviation list (P20L406-P21L418).

Minor revision from authors.

1. We revised the structure of sialic acid in Fig. S4.
2. We revised the “Thus” to “Collectively” in P8L131.
3. We added “In addition, ” in P12L230, P12L240
4. We revised the typo; from “gnomic” to “genomic” in P19L390.
5. We added the explanation of survival curve in P33L639.
6. We added the sentence regarding to HA concentration used in microarray in Fig. S6
P43L721-722.
7. We added the explanation of bar charts in P46L743-744.

Reviewer #1 (Remarks to the Author):

Together with a junior colleague, I carefully reviewed the revised version of the manuscript. We found that the authors have substantially improved the readability of the text, making the manuscript clearer and more accessible. All of the points we previously raised have been addressed thoroughly and to our full satisfaction.

Furthermore, we noted that the authors have also responded exceptionally well to the concerns raised by the other referees, demonstrating a thorough and thoughtful approach to all feedback received.

Overall, we believe that the revised version is now ready for acceptance for publication.

Reviewer #2 (Remarks to the Author):

Reviewer #3 (Remarks to the Author):

The authors addressed all points raised in my first review largely sufficiently.

Two main points (term “hypervirulence” and describing generation and characterization of recombinant L-PTC/B) remain to become resolved, see below.

L82-3: not precise enough: Is the dose related to 150 kDa BoNT or the complex? For serotype F, authors list “M-PTC serotype F1: 2.4-3.6”. SI unit is missing and why here M-PTC instead of BoNT?

In this manuscript, BoNT means BoNT alone, not the complex.

We added SI unit to the LD50 of F1 toxin. And here we cited the value of M-PTC because they referred to the toxicity of M-PTC serotype F1. For clarity, we only cite the LD50 of BoNT, not M-PTC (L82-83), and added a new reference (ref.14).

L84-5: “...the L-PTC from serotype B1-Okra (L-PTC/BOkra, termed hypervirulent) has at least 80-fold or 20-fold (Fig. 1a) higher oral toxicity than that from serotype A1-62A...”. Rewrite: “L-PTC/BOkra has 20-80-fold higher oral toxicity than that...”.

We have revised it (L84).

Again, virulence is defined as the strength of pathogenicity of a virus or bacteria. *C. botulinum*

is pathogenic due to the expression of BoNT. The potency/specific toxicity of the BoNT in conjunction with its expression levels define the virulence of the *C. botulinum* strain. I strongly recommend to replace hypervirulence by increased oral potency.

In response to the reviewer's comment, we have replaced the term "hypervirulent" with "hyper-oral-toxic" throughout the manuscript. (e.g., L51-52, L84-86, L89, L156, L202, L237, L239, L244-245, L590)

Serotype A-62A is still a potent toxin, although its oral toxicity is 20-80 fold lower than that of serotype B-Okra. To avoid implying that serotype A-62A is not orally potent, we used the term "hyper" and "non-hyper" rather than "higher" and "lower" toxicity.

Apart from the term, what is your definition for "hypervirulence"? Do you set an absolute minimum oral toxicity in $\mu\text{g}/\text{kg}$ bodyweight of 150 kDa BoNT and everything above is "hypervirulent"?

Currently, no formal criteria have been established. However, the oral toxicity of serotypes A-62A and B-Okra differs significantly as shown Fig.1 (>20-fold) and ref. 1 (~80-fold). We simply propose this classification based on the observed differences. Further studies involving additional serotypes are needed to define appropriate classification criteria.

And is then M-PTC/F "hypovirulent"?

According to ref.1 (Sakaguchi. Pharmacol Ther 1982; doi:10.1016/0163-7258(82)90061-4), purified M-PTC/E and M-PTC/F are hypovirulent (or non-hyper-oral-toxic). In contrast, Gao, et al. (ref.3) recently reported that OrfX proteins enhance the oral toxicity of serotype E1 (M-PTC/E). However, its potency remains much lower than that of serotype B-Okra and is similar to or even lower than serotype A-62A.

L121: Change "The M-PTC from neither toxin bound to PGM" to "...from neither serotype..."

We have revised it (L120).

L125-7: Cite "A nut-and-bolt assembly of the bimodular large progenitor botulinum neurotoxin complex" Lam et al. accepted in Science Advances which showed inter-serotype combination of NTNHA and HA70.

We thank the reviewer for the helpful information. However, as the referenced paper has not yet been published, it is not currently accessible to the public.

L131: I checked the Toxins manuscript describing generation and characterization of recombinant L-PTC/B. The authors need to make sure that the Toxins manuscript will be

published prior to this manuscript otherwise the Material & Method section of this manuscript is insufficient by lacking the generation and characterization of recombinant L-PTC/B which is an integral part of the study.

We have submitted the manuscript to Toxins and simultaneously deposited the manuscript to an MDPI preprint server (doi: 10.20944/preprints202507.0898.v1). We have updated the citation (L480-482).

L233: Change “pathogenicity of BoNT” to “mechanism of intoxication by BoNT”

We have revised it (L235).

Reviewer #4 (Remarks to the Author):

Clearly, the manuscript is much improved in all respects and reports important results. Still there are aspects that can be further improved.

With respect to the crystallographic analysis, I am basing my judgement on Table 2, the text in the manuscript and the validation reports as the PDB entries are not yet released. Both structures are significantly improved. The low-resolution structure (PDB ID: 9UG5) still has many weak aspects, among others the very high Rmerge (10.5%) and B-factors. It is quite possible that there is not much more to achieve with this data set, although the authors might like to double check if the space group is potentially only pseudo-orthorhombic (try e.g. P21). We appreciate the reviewer for their careful evaluation and valuable suggestion regarding the space group assignment.

In response, we systematically tested a range of alternative space groups: P2, P21, P222, P2122, etc. However, none of these settings resulted in improved refinement statistics or better map quality. The original space group assignment remains the most appropriate based on overall data quality, refinement behavior, and crystal packing.

Good is the electron density fit of the carbohydrate ligand. However, it is very surprising that the authors chose to build water molecules into 3 Å density. This is not warranted until approx. 2.5 Å resolution. The authors might like to double check if these are ions.

Upon re-examination of the electron density maps, we have removed water molecules modeled at over 2.5 r.m.s.d. of the Fo-Fc map. And we have updated the refinement statistics sheet (Supplementary Table 1) and attached the refined models.

– The 2.2 Å structure (PDB ID: 9UG6), due to its higher resolution, is of better overall quality, however, here the electron density for the carbohydrate ligand is much less convincing, with large patches of negative difference density indicating errors. I wonder at which state the

authors built in the ligand (as this is not commented)? Was it at the very end of the refinement or early on and thus not unbiased? The authors might like to remove the ligand, run simulated annealing and then add back the ligand into unbiased density or use Polder maps.

The ligand was initially modeled early in the refinement process after molecular replacement. At a later stage, we removed it, conducted refinement including simulated annealing, but observed no significant improvement in electron density. However, further refinement of the SL6 model using REFMAC reduced the negative difference density. We have updated the refinement statistics sheet (Supplementary Table 1) and attached the refined models.

We believe the poor ligand density in the higher-resolution structure (PDB ID: 9UG6) may reflect the fact that ligand binding is unfavorable for crystal packing in this particular form. In support of this, the ligand-binding site coincides with a region involved in crystal contacts, suggesting that ligand binding might introduce local disorder or steric hindrance detrimental to lattice stability. In addition, among the three molecules in the asymmetric unit, the ligand binding is observed only in one or two of them.

These binding modes are consistent with previously reported HA3/A-SL complex structures (PDB 4LO6). Since they are not directly related to the novel aspects discussed in this study, we do not discuss this part in the manuscript.

For this entry, I observed a discrepancy between the Rmerge in Table 2 (4.7%) and in the validation report (6%). This should be clarified.

We have re-checked our data and confirm that the Rmerge value listed in Supplementary Table 2 (4.7%) is correct. The slightly higher value (6.0%) in the validation report likely arises from differences in the resolution range or inclusion of unmerged outer-shell data in the automated analysis. The Rmerge value of Supplementary Table 2 (4.7%) was calculated from 41.44–3.00, whereas that of the validation report (6.0%) was calculated from 44.32–3.00.

The difference between R and Rfree (7.4%) is very high and it should be possible to improve this significantly given the data quality. (While they are at it, the authors might also like to improve their previous structure, 3WIN, which they used for molecular replacement).

Regarding the R–Rfree gap (7.4%), we agree that this is higher than ideal. However, despite multiple rounds of refinement and model optimization, we were not able to significantly reduce this gap. We believe this reflects the relatively weak map quality for this dataset, which limits further model improvement, despite the nominal resolution. We consider the current statistics to be within an acceptable range given these constraints.

We also appreciate the suggestion to revisit our earlier structure (PDB ID: 3WIN). While it served adequately as a molecular replacement model, we will explore the possibility of updating this entry in a future revision.

– Apart from this, I suggest the following improvements of Table 2: Give 3-4 digits for the wavelength, but only one digit for unit cell parameters (except crystallographic angle), completeness, Ramachandran distribution and B-factors.

We have revised it (Supplementary Table 1).

Add unit (\AA^2) for B-factors.

We have added it (Supplementary Table 1).

The Methods section could also be further improved: Report the refinement strategy (e.g. when were ligands and water molecules added and according to which criteria);

We added the refinement strategy in the Methods section as follows (L373-377):

“Structure refinements were carried out using and Coot⁵² and REFMAC5⁵³. Initial refinement was performed using rigid-body refinement followed by restrained refinement with isotropic B-factors. After initial refinement, carbohydrate ligands were identified based on positive electron density peaks in the Fo-Fc maps. In the final stage of refinements, waters were automatically added to the models and manually curated.”

clarify what exactly the cryo-protectant contained (also buffer? And I assume the respective sialyllactose ligands 3SL and 6SL);

We have added the exact cryo-protectant information (L369-370).

and remove the redundancy at the start of the section (e.g., shorten to “...(Tokyo Chemical Industry). Crystallization was performed using Cryschem plates...and the droplets consisted of 3 ul protein + ligand and 3 ul reservoir solution containing xxx.”

We have revised it (L365-366).

Overall, I would like to commend the authors for improving the readability of the manuscript. Still, with my background, the first part of the manuscript was challenging to read. For example, I would benefit from additional details in the legend to Figure 1, explaining early on that vehicle consists of sterile water and what GP2 is.

In this experiment, NAC was dissolved in sterile water. Therefore, we gavaged mice with sterile water alone as vehicle control. GP2 is glycoprotein 2, a marker protein for M cells. We added the

explanation to the manuscript and the legend (vehicle, L301, L605; GP2, L598). Also, GP2 was added to the abbreviation list (L421).

Given that fucosylation is so important, I also recommend to add more references in the Discussion (p. 12, lines 230-232). For example, many people do not even know about the importance of fucosylated cholera toxin receptors (as reported by Wands et al., Cervin et al and many papers from the Krenzel lab). By the way, secretors are also protected from cholera (line 248).

We added references to each pathogen and cholera protection (L233-234).

Finally, to drive the main points home, the authors may consider adding as Figure 5 a cartoon drawing of the main findings.

We added the cartoon drawing to Supplementary Figure 13.

Minor points for correction:

- Introduction:

o Line 68: Is there a consequence of combination with OrfX2 protein? (extension of the sentence could be helpful as it is difficult to read).

OrfX2 binds to M-PTC and forms an M-PTC–OrfX2 complex. We have extended the sentence as follow: “*which can be further combined with hemagglutinin (HA) to form L-PTCs or with OrfX2 protein to form M-PTC–OrfX2 complex*” (L68).

o HA: I was mainly aware of HA in the influenza virus. Add and additional explanatory sentence?

While the term “hemagglutinin (HA)” is also used in the context of influenza, measles, and mumps viruses, the HA mentioned in this manuscript refers specifically to the hemagglutinin of botulinum toxin complex. Since other types of HA are not discussed in this manuscript, we believe no additional explanation is necessary in the manuscript.

o Line 85: “at least 80-fold or 20-fold” sounds strange – rephrase to “more than 20-fold and give reference and call figure thereafter?”

We have revised the sentence from “*at least 80-fold or 20-fold*” to “*at least 20–80-fold*” (L84).

- Line 94: How was it shown that L-PTC/BOKra underwent endocytosis? (not clear to me, but probably obvious to cell biologists)

The puncta in the lower panel of Fig. 1b represent the endocytosed protein, but, as the reviewer

pointed out, this may not be clear. To address this, we added a reference to Supplementary Fig. 1 in addition to Fig. 1b (L94).

- Line 130: Sentence strange (equivalent?)

We have revised the sentence from

“we confirmed that each rL-PTC carries equivalent the BoNT activity (Fig. 2e) and intact HA activities” to “we confirmed that each rL-PTC carries equivalent BoNT activity (Fig. 2e) and retains intact HA activities” (L128-130).

- Line 167: Should be Fig. 3 (not S5)

We intended to refer to kinds of carbohydrates and lectins, not the results. For clarity, we have revised the sentence “inhibition/competition ELISA (Fig. S5)” to “inhibition/competition ELISA with carbohydrates/lectins (Supplementary Fig. 5)”.

- Methods: add where the sugars and lectins were purchased from (only done for some of the compounds)

We added the information (L341-344, L349-350).

- List abbreviations alphabetically (easier to find)

We sorted the abbreviations alphabetically (L416-428).

- Decapitalize references

We have revised it (L441-569).

- Figures:

o Fig. 1: Lines 578-580 (d should also apply to f and h (log-rank test) to a.

We have revised it. (Fig. 1a and L612-614)

Add specification of vehicle and GP2.

We specified these. (L324, L421, L598, L605)

o Fig. 2: Consider referring to Fig. S4 in f.

We referred to Supplementary Fig. 4 in the legend of Fig. 2f (L629-630).

o Fig. 3: It could be helpful to refer to Fig. S5 in the legend.

We have already referred to Supplementary Fig. 5 in the legend (L649-650).

Line 616: It should be Fig. S5 (not 5) and (43) should be deleted.

We have revised it (L650).

Line 620: Not completely clear – refer to crystal structures 4LO2 and 4OUJ and docked complex with fucose added (see Fig. S7).

We added the following sentence for clarity:

“Crystal structures of HAI/A62A-Lac (PDB ID 4LO2) and HAI/BOkra-Lac (PDB ID 4OUJ) are superimposed with 2FL” (L655-656).

o Fig. 4c: Consider Making symbol for Fut2-/- half red and half blue.

We appreciate the suggestion. However, we prefer to keep the current color scheme in Fig. 4c as it maintains consistency with the rest of the figures and preserves clarity for readers already familiar with the symbol.

o Fig. S2: Line 678 (T ->t). Line 682 (B -> b).

We revise from T to t (T->t) in L714, while L718 is correct (abbreviation; B, BSA).

And: I am not quite sure if it is okay to normalize to BSA-coated wells, giving >100% in permeability in c?

Regarding the reviewer’s concern about normalizing to BSA-coated wells, which results in values exceeding 100% in panel c of Supplementary Fig. 2, we believe this is acceptable. We consider that the mucin-coated wells exhibited lower non-specific binding of the toxin, thereby resulting in relatively higher apparent permeability values (>100%).

o Fig. S5: Much appreciated! – Can be further improved formally, by using convention of small caps for D/L.

The D and L are already formatted in small caps, although we understand they may be difficult to distinguish in this format (L739-740).

Line 708: Legend refers to Gal-beta 1,3 GalNAc specificity of PNA, but figure shows Gal-beta 1,3 GlcNAc. SBA: GalNAc and Gal-specific. ConA: Man- and Glc-specific. SSA: misleading, as it is not GalNAc-specific (add Sia). Fig. S9 Refer to SNFG symbol nomenclature (<https://www.ncbi.nlm.nih.gov/glycans/snfg.html>).

PNA recognizes Gal-beta 1,3-GalNAc. We have corrected Supplementary Fig. 5c according to the SNFG symbol nomenclature and revised the figure legend (L674, L745-753, L788).

o Fig. S9: Line 741: Maybe better write “Recombinant wild-type a1,2 fucosidase AfcA (WT) to avoid the impression that these are two difference versions of the enzyme?”

We have revised it (L777).

o Fig. S10: revise nul -> null

We have revised it (L785).

Minor corrections from author:

-In Supplementary Fig. 2b, the lane labels of anti-L-PTC/B were incorrect, and we have revised it: from “B P M B P M” to “B P M B P M”. (B, BSA; P, PGM; M, MIM)

-Format change: e.g., Fig. S1 -> Supplementary Fig. 1 (throughout the manuscript)

-In several places, the amino acid numbers of HA were incorrect. We have revised these; e.g., from “*His282^{A-62A}/Asn281^{B-Okra}*” to “*His281^{A-62A}/Asn281^{B-Okra}*”. (L153-154, L179, L189, L190, L724, L726, L729)

-In Supplementary Fig. 5c, the cartoon of the PNA-recognizing carbohydrate was incorrect (Gal-beta1,3-GlcNAc). We have revised it to the correct structure, Gal-beta1,3-GalNAc.

Typo:

“intraperitoneal” -> “intra~~g~~astric” (L128)

“enter” -> “entered” (L238)

“0,056” -> “0.056” (L254)

“L-PTC/A62A and L-PTC/BOkra” -> “L-PTC/A^{62A} and L-PTC/B^{Okra}” (L283)

“statics” -> “statistics” (L378)

“pannel” -> “panel” (L672)

“alignment” -> “alignments” (L723)

“Alexa Floure” -> “Alexa Fluor” (L809)

Reviewer #4 (Remarks to the Author):

I am pleased with the authors' revision and only have three small comments left:

a) on p. 13 l. 250 (Discussion), it should be [infection with] Vibrio cholerae - not Cholera

We have revised it. (P12L252 in main manuscript)

b) typo in the legend of Fig. S5 (p. 42, l. 753): Samubucus -> Sambucus (remove u)

We have revised it. (P17 L355 in main manuscript, P9L72 in Supplementary Information)

c) I recommend that the beautiful Fig. S13 is moved to the main manuscript.

We have moved Fig. S13 to the main manuscript as Fig. 5.

In addition, as encouraged by the journal, I shared the manuscript with a junior researcher, who had the following comments:

The manuscript presents very interesting results about the role of mucin fucosylation in BoNT intoxication. The work is supported by many different experiments, including cell biology, binding assays, toxicity studies, and even some structural biology. It was nice to read. I particularly liked supplementary figure 13 showing the model of how the toxins pass through the mucin and enter the host. I think this figure should be moved to the main text, as it summarizes the findings quite well.

We have moved Fig. S13 to the main manuscript as Fig. 5.

I also liked that the authors included some structures of the toxins bound to sugars, but I would have liked to see more discussion about why HA1/A62A can accommodate the fucosylated sugar better than HA1/BOkra.

We appreciate this good point. The His residue may be sufficiently flexible to accommodate the fucosylation, whereas Asn appears more constrained. While steric hindrance may contribute to the observed difference between HA1/A-62A and HA1/B-Okra in accommodating the fucosylated sugar, the available data are insufficient to draw a definitive conclusion. Further structural analyses will be needed to clarify the underlying mechanism. Instead, we have added a detailed description of the current situation in the manuscript as follows:

P10L179-180, "...extended toward the His281A-62A/Asn282B-Okra residues, suggesting that the

Asn282B-Okra does not accommodate the fucose-extension.

P10L185-186, “*This indicates that fucosylation neither facilitates nor impairs HA1/A62A binding to glycans.*”

I have some additional suggestions that could further improve the paper:

-In the abstract (line 50), the authors could specify that B-Okra refers to a serotype.

We have revised “*Clostridium botulinum B-Okra*” to “*Clostridium botulinum serotype B-Okra*”. (P3L50)

-It would be nice to add a figure (early in the text) showing the general architecture of the L-PTC complexes (similar to the one shown in Fig2.d for the toxin chimeras). They are described at the beginning of the introduction, but a figure can help the readers visualize the individual components and how they are arranged, making the paper easier to follow from the start.

We have added the schematic diagram of L-PTC to Fig. 1a.

-The introduction ends in the knowledge gap, but it could be a good idea to add a couple of sentences about the research question and the main findings. This is already mentioned in the abstract, but it could be stated again in this section.

We have added a brief sentence about our findings at the end of the introduction section. (P5L85-87)

“Here, we show that the mucus fucosylation determines the entry routes of botulinum toxin complexes in the gut and their oral toxicities.”

-In the methods section ‘Toxins and neurotoxin-associated proteins (NAPs)’, maybe specify that the NAP components were produced separately in E. coli (and not co-expressed).

We have revised the method and emphasized that the toxin was not expressed in *E. coli* as follows:

“For biosafety considerations, active bont genes, in any form, were never expressed in Escherichia coli. Recombinant NAPs were produced by E. coli as previously described”. (P13L277-279)

-In the results, it was not made clear which structures were obtained in this work. At least in Figure 2.f, line 627, add ‘(this work)’ after the new structure is mentioned.

We have revised the legend of Fig. 2f to include the term “this work”. (P28L631)

“... of HA (HA1/A^{62A}-Lac: PDB ID 4LO2²²; HA1/B^{Okra}-Lac: 4OUJ²¹; HA3/A^{62A}-SL6: 4LO5²²; HA3/B^{Okra}-SL6: 9UG6, this work)”

Minor points for correction:

-Line 114: 'mutations' is repeated twice.

We have revised it. (P7L144)

-Line 145: 'WT vs. 619 mutants' sounds strange. Write either 'residue 619 mutants', or Arg/Lys 619 mutants to make it clearer.

We have revised “WT vs. 619 mutants” to “WT vs. Arg/Lys 619 mutants”. (P7L145)

-Line 322: write full name of reagent: 1x cOmplete™ Protease Inhibitor Cocktail

We have revised it. (P15L328-329)

-Line 644: make the letter 'b' in the figure legend bold.

We have made the letter “b” bold in the legend of Fig. 3b. (P29L649)

-Lines 626, 627 and 629: α2-6-sialyllactose is abbreviated as 'SL6', but in the rest of the text it is '6SL'. Change to 6SL for consistency.

We have unified the abbreviation to “3SL/6SL” throughout the manuscript and figures.